# The importance of Aitken mode aerosol particles for cloud sustenance in the summertime high Arctic: A simulation study supported by observational data

Ines Bulatovic[1,2], Adele L. Igel[3], Caroline Leck[1,2], Jost Heintzenberg[1,5], Ilona Riipinen[4,2] and Annica M. L. Ekman[1,2]

[1]Department of Meteorology, Stockholm University, Stockholm, 106 91, Sweden
[2]Bolin Centre for Climate Research, Stockholm, 106 91, Sweden
[3]Department of Land, Air and Water Resources, University of California, Davis, Davis, CA 95616, California
[4]Department of Environmental Science (ACES), Stockholm University, Stockholm, 106 91, Sweden
[5]Leibniz Institute for Tropospheric Research, Permoserstr. 14, Leipzig, 04318, Germany

*Correspondence to*: Ines Bulatovic (ines.bulatovic@misu.su.se)

**Abstract.** The potential importance of Aitken mode particles (diameters ~25–80 nm) for stratiform mixed-phase clouds in the summertime high Arctic (> 80° N) has been investigated using two large-eddy simulation models. We find that in both models Aitken mode particles significantly affect the simulated microphysical and radiative properties of the cloud and can help sustain the cloud when accumulation mode concentrations are low (<10–20 cm$^{-3}$), even when the particles have low hygroscopicity (hygroscopicity parameter $\kappa$=0.1). However, the influence of the Aitken mode decreases if the overall liquid water content of the cloud is low, either due to a higher ice fraction or due to low radiative cooling rates. An analysis of the simulated supersaturation (ss) statistics shows that the ss frequently reaches 0.5% and sometimes even exceeds 1%, which confirms that Aitken mode particles can be activated. The modelling results are in qualitative agreement with observations of the Hoppel minimum obtained from four different expeditions in the high Arctic. Our findings highlight the importance of better understanding Aitken mode particle formation, chemical properties and emissions, in particular in clean environments such as the high Arctic.

## 1 Introduction

The Arctic region is experiencing a rapid increase in surface temperature that is substantially larger than the global average increase (Holland and Bitz, 2003; Hartmann et al., 2013). The enhanced Arctic warming, known as the Arctic amplification, is a result of local drivers and feedbacks (e.g., local aerosol sources, cloud and ice-albedo feedbacks, lapse-rate feedback; Serreze and Barry, 2011; Stuecker et al., 2018; Stjern et al., 2019) as well as remote forcings (which modify heat and moisture transport from lower latitudes). Changes in the dynamical and microphysical properties of clouds are central to local feedbacks (Curry et al., 1996; Garrett et al., 2009; Kay et al., 2011) due to the strong impact of the clouds on the surface energy budget (Curry and Ebert, 1992; Shupe and Intrieri, 2004) and subsequent sea-ice growth (Kay and Gettelman, 2009; Kay et al., 2011; Tjernström et al., 2015).

Low-level, stratiform mixed-phase (SMP) clouds are persistent and frequent in the Arctic (Shupe et al., 2006; Shupe et al., 2013). Despite the presence of liquid and ice in the same volume and a continuous sink of the liquid phase through ice growth and precipitation, these clouds may persist for several days (Shupe et al., 2006). A layer of liquid is typically present at the top of SMP clouds (e.g., Shupe et al., 2006; Morrison et al., 2012). Maintenance of this layer is critical for sustaining longwave emission and ensuing cooling at the cloud top (e.g., Persson et al, 2017; Dimitrelos et al., 2020), which enhances a buoyancy-driven turbulent mixing in a layer within and below the cloud (e.g., Tjernström et al., 2005). The turbulence further increases cloud liquid water as strong overturning means strong updrafts that allow efficient condensation of water vapor onto cloud droplets. It also leads to higher entrainment rates at the cloud top (e.g., Tjernström, 2007). A peculiar feature of the Arctic region is that the specific humidity frequently increases over the inversion layer that caps the low-level SMP clouds (Sedlar et al., 2012; Shupe et al., 2013). Entrainment may thus bring more vapor into the cloud and moisten the boundary layer (Solomon et al., 2011; Tjernström et al., 2012; Solomon et al., 2014; Loewe et al., 2017). This and other conditions specific for boundary layers in the Arctic allow liquid water to persist, and thereby also prevent quick cloud glaciation, despite an opposing effect of ice growth within the cloud (e.g., Morrison et al., 2012).

In the high Arctic (north of 80° N), SMP clouds have a net warming effect on the surface energy budget during most of the year. Due to a low amount of solar radiation in this region, the warming induced by cloud longwave emission towards the surface is generally larger than the cooling effect due to reflection of solar radiation.

However, during the peak-melt season at the end of the summer, high Arctic SMP clouds can have a net cooling effect on the surface energy budget and thereby influence the surface temperature and timing of the autumn freeze-up (e.g., Intrieri et al., 2002; Shupe and Intrieri, 2004; Tjernström et al., 2014). Aerosol particles influence the radiative effect of clouds as they act as cloud condensation nuclei (CCN) and ice nucleating particles (INP) and affect the microphysical and optical properties of clouds (referred to as aerosol indirect effects). Due to the generally pristine conditions in the high Arctic (e.g., Bigg and Leck, 2001), the clouds in this region can be particularly sensitive to the aerosol perturbations. Previous modelling studies of high Arctic clouds (e.g., Birch et al., 2012; Loewe et al., 2017; Stevens et al., 2018) have indeed shown that the cloud liquid water content and cloud radiative properties are highly sensitive to the concentration of CCN at low CCN (referred to as the tenuous cloud regime). Therefore, there is a need to better understand sources and sinks of high Arctic aerosols, the seasonal variability as well as their chemical composition, physical characteristics and potential effects on cloud formation. However, due to the harsh conditions, measurements are sparse and generally limited to summertime. Overall, the annual cycle of aerosol particle concentrations in the whole Arctic is characterized by transport of anthropogenic emissions from lower latitudes during the winter season with a peak in April (known as the Arctic haze), relatively pristine conditions during the summertime, and a minimum in the fall (Heintzenberg and Leck, 1994; Tunved et al. 2013; Freud et al., 2017). When long-range transport of aerosols over the pack ice is small, as in summer, the surface number concentrations of accumulation mode particles (sizes typically 80–500 nm; Covert et al., 1996) in the high Arctic are generally below 100 cm-3 and occasionally below 1 cm-3 (Bigg et al., 1996; Mauritsen et al., 2011; Heintzenberg et al., 2015; Leck and Svensson, 2015). During this time of the year, marine biological activity can provide a source of small, airborne particles, adding to the mass and number of Aitken mode particles (sizes typically 25–80 nm; Covert et al., 1996) (Leck and Bigg, 2005a; Heintzenberg and Leck, 2012; Karl et al., 2013; Heintzenberg et al., 2015).

The ability of an aerosol particle to act as a CCN depends upon multiple factors, such as its size and chemical composition (Köhler, 1936), surface tension (e.g., Ovadnevaite et al., 2017) and the ambient relative humidity (e.g., Rastak et al., 2017). A larger maximum supersaturation within an air parcel allows smaller and less hygroscopic particles potentially to act as CCN (Köhler, 1936; Petters and Kreidenweis, 2007). On the other hand, the maximum supersaturation is also dependent on the relative abundance of particles, in particular the number of water soluble accumulation or coarse mode particles as they easily act as CCN and subsequently take up water when they grow (e.g., Ghan et al., 1997). Therefore, typical CCN sizes differ among environments with different aerosol size distributions, composition and supersaturation values (Seinfeld and Pandis, 2006). In general, water soluble particles within the accumulation mode constitute the largest source of atmospheric CCN (Seinfeld and Pandis, 2006). However, in the summertime Arctic, a relatively low condensation sink of water vapor due to the low number of accumulation mode particles may lead to relatively large maximum supersaturations that could allow Aitken mode particles (that are typically more abundant) to act as CCN. The previous studies that have analyzed the effect of CCN in the tenuous cloud regime have not distinguished between the aerosol particles of different sizes and properties that can have different impacts on clouds. Recent observations for the summertime Arctic region south of the ice edge, have suggested that particles with diameters below 50 nm can be CCN-active (Willis et al., 2016; Kecorius et al., 2019; Koike at al., 2019). However, these analyses were not performed for the high Arctic and they did not explicitly investigate the relation between Aitken particles and cloud properties or cloud sustenance. Instead, they focused on the correlation between aerosol particles and cloud droplets. Furthermore, the environment over the pack ice is unique with fewer aerosol sources and different surface conditions compared to the open ocean (e.g., Leck and Svensson, 2015). This could lead to an even stronger influence of Aitken mode particles than south of the ice edge. Model simulations by Christiansen et al. (2020) have indicated that Aitken mode particles can influence high Arctic cloud properties, but their simulations only considered extreme conditions with no accumulation mode aerosols present in the atmosphere.

Summarizing, we know that high Arctic summertime SMP clouds over the pack ice are governed by a complex interplay between dynamics, cloud microphysics and aerosols and that they strongly influence climate, however, there are still many uncertainties regarding these clouds. One knowledge gap is if and under which conditions Aitken mode particles become CCN-active in this environment and how these particles then may affect the microphysical properties of the clouds. In this study, we therefore employ two different large-eddy simulation (LES) models to simulate a relatively long-lived summertime cloud observed in the high Arctic during the ASCOS campaign (Tjernström et al., 2014). During the campaign, measurements often showed low concentrations of accumulation mode particles while the concentration of Aitken mode particles was relatively high (Leck and Svensson, 2015). We initialize the models with a range of aerosol size distributions and explore if Aitken mode particles can help sustain the cloud or if only accumulation mode aerosols control cloud properties (i.e. cloud droplet, rain and ice mixing ratios), even at low accumulation mode concentrations. We also analyze the maximum supersaturations simulated by the two models and calculate the corresponding threshold diameters of aerosol activation. The engagement of two different models allows us to evaluate if the results are dependent on the details of a specific model or if we can draw more general conclusions. Finally, we introduce the statistics of the aerosol size distributions (Heintzenberg and Leck, 2012) observed during the summers of four different high Arctic

campaigns that took place in 1991, 1996, 2001 and 2008 year (Leck et al., 1996; Leck et al., 2001; Leck et al., 2004; Tjernstöm et al., 2014) and compare them with the simulated results. The general conclusions are provided at the end of the study.


## 2 Method

### 2.1 Models

The simulations were performed using two models. MIMICA (the MISU MIT Cloud-Aerosol Model) is an LES model that has been successfully used for simulating Arctic mixed-phase clouds (e.g., Ovchinnikov et al., 2014; Savre et al., 2015; Igel et al., 2017; Stevens et al., 2018, Christiansen et al., 2020). The model solves the equations for a non-hydrostatic, anelastic atmospheric system; a full description of the model can be found in Savre et al. (2014). A two-moment bulk microphysics scheme (Seifert and Beheng, 2001) is used to calculate the prognostic
variables (i.e. mass mixing ratio and number concentration) of five different hydrometeor types considered, namely cloud droplets, raindrops, cloud ice, graupel and snow. All hydrometeor categories have mass distributions in the form of regular gamma functions. Autoconversion and self-collection of liquid particles are also calculated as described in Seifert and Beheng (2001). A pseudo-analytic method is used to model the supersaturation, with the integration of condensation/evaporation at the model time step of ~2 s (Morrison and Grabowski, 2008). The
terminal fall speed of the hydrometeors is calculated using a simple power law of the diameter of the particle, which determines the wet deposition (Pruppacher and Klett, 1997). To represent an aerosol population of different particle sizes and chemical compositions, MIMICA includes a two-moment aerosol module (Ekman et al., 2006). All aerosol modes are described with lognormal distributions. In the model, aerosols can act as cloud condensation nuclei following kappa-Köhler theory (Petters and Kreidenweis, 2007), but not as ice nuclei. The number
concentration of ice crystals is prescribed and kept quasi-constant during the simulations (Ovchinnikov et al., 2011, 2014). This parameterization mimics immersion freezing, i.e. ice can only form if there is supercooled water present. Aggregation of ice crystals is permitted in the model. The radiative transfer is calculated following a four-stream radiative transfer solver (Fu and Liou, 1993), which includes 6 bands for shortwave and 12 bands for longwave radiation.
The second model used is the Regional Atmospheric Modeling System (RAMS; Cotton et al., 2003), which has also been successfully used in studies of Arctic mixed-phase clouds (e.g., Avramov and Harrington, 2010; Ovchinnikov et al., 2014). RAMS is a flexible model that is most commonly used for cloud-resolving and large eddy simulations. It uses a two-moment bin-emulating bulk microphysics scheme to predict the mass and number mixing ratios of liquid and ice hydrometeor species (Meyers et al., 1997; Saleeby and van den Heever, 2013). In
this study, six species are used, namely, cloud droplets, raindrops, cloud ice, snow, graupel and hail. RAMS typically uses two cloud ice species, but only one was used in this study as was done in MIMICA. Collision-coalescence of liquid drops is done through the use of lookup tables that are generated by solving the stochastic collection equation (Feingold et al., 1997). Condensation depends explicitly on the hydrometeor properties and allows for supersaturation at the end of the time step of 1.5 s (Walko et al., 2000). The terminal fall speed of the
hydrometeors is calculated based on piecewise power laws (Mitchell, 1996). RAMS also includes a user-defined number of lognormal aerosol distributions. Aerosol particles act as CCN and are activated using additional look-up tables generated from an offline parcel model based on kappa-Köhler theory (Saleeby and van den Heever, 2013). Aggregation of ice crystals is permitted. Radiative transfer is calculated following Harrington (1997).

### 2.2 Overview of the simulated case: The ASCOS campaign

The Arctic Summer Cloud Ocean Study (ASCOS) campaign took place in the summer of 2008 onboard the Swedish icebreaker *Oden*, and included a three-week ice drift with enhanced meteorology measurements when *Oden* was anchored to a large ice floe slightly north of 87° N. A full description of the expedition can be found in
Tjernström et al. (2014). This campaign has so far been one of the most extensive studies in the central Arctic focusing on the atmosphere, clouds and aerosol properties and their linkages to the microbiological life in the upper ocean. To investigate a case with a quasi-steady-state cloud regime, the simulations are based on a period that was characterized by a persistent, low-level SMP cloud observed from 18 UTC 30 August to 12 UTC 31 August 2008. The period represents one of the last days of the ice drift episode, which took place from 12 August
to 2 September 2008. During this period, the number concentration of accumulation mode particles was relatively low (Leck and Svensson, 2015). Therefore, a change in the aerosol population could be particularly important for inducing cloud perturbations that may affect the surface energy budget.
During the ice drift, radiosondes were launched from the ice surface every 6 h and provided profiles of thermodynamic properties (e.g., pressure, temperature, relative humidity) and wind speeds (cf. Figure 1). The one
from 05:35 UTC 31 August 2008 was representative of the conditions observed during the stratocumulus period

and used to initialize the simulations. Cloud properties and thermodynamic characteristics of the atmosphere were monitored with surface-based remote sensing instruments (Shupe et al., 2013). The cloud base and cloud top were nearly constant during the cloud lifetime (500 and 1000 m, respectively). Retrievals of liquid water path (LWP) were made from the 23 and 30 GHz microwave radiometer measurements (Sedlar and Shupe, 2014). The combination of different sensors was used to estimate ice water path (IWP; Shupe et al., 2008). The observed LWP uncertainty was around ~25 g m$^{-2}$ while the uncertainty in the IWP was about a factor of two (Shupe et al., 2008; Birch et al., 2012). A CCN counter that was situated on *Oden*, at 25 m above the sea surface, measured a mean CCN concentration of about 25 cm$^{-3}$ at a supersaturation of 0.2 % during the period of the ice drift (Martin et al., 2011; Leck and Svensson, 2015). The ship and the inlets were facing the wind so that local pollution from the ship was avoided. Furthermore, a pollution controller was used to prevent direct contamination from the ship and the main pumps were turned off whenever the conditions for a clean environment were not completely satisfied (details on the pollution control system can be found in Leck et al., 2001 and in Tjernström et al., 2014). Since the surface boundary layer typically was decoupled from the turbulent layer associated with a cloud (Tjernström et al., 2012), it is however not certain if the CCN concentrations measured at the ship were representative for the cloud layer (cf. also observed vertical profiles of particle concentrations in Igel et al., 2017, Figure 1).

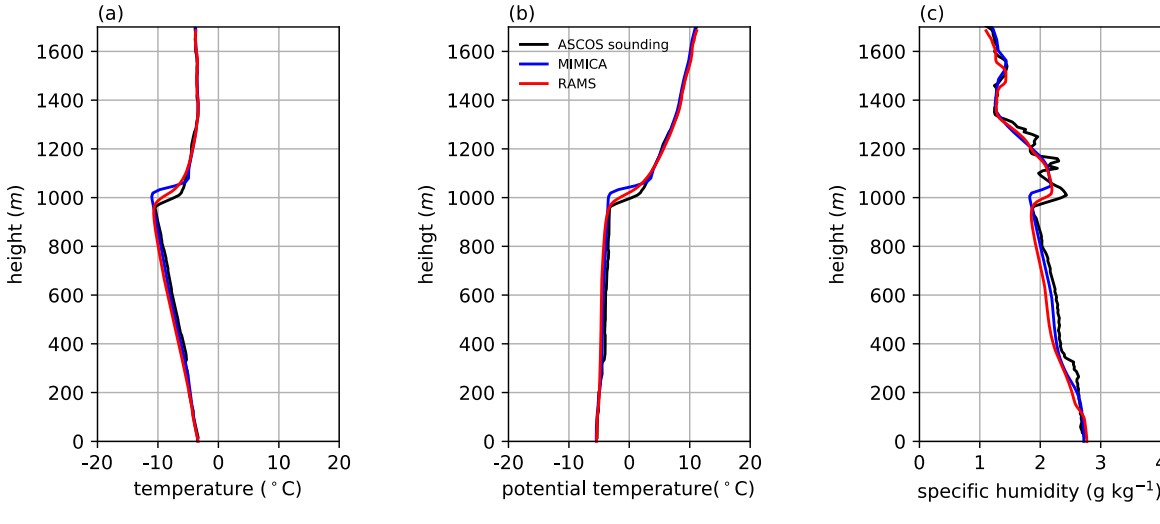

**Figure 1: Radiosonde observations from 05:35 UTC 31 August 2008 of (a) absolute temperature (b) potential temperature and (c) specific humidity. The corresponding simulated profiles from MIMICA and RAMS are presented for the middle of the simulation period (6h of simulation time).**

**2.3 Simulation setup**

We performed simulations with different prescribed aerosol size distributions to investigate the influence of Aitken mode particles on cloud properties, and the dependence of this influence on the background concentration of accumulation mode aerosols. For both models, there is no sink or source of aerosols within the model domain during the simulations. We conduct two sets of simulations with two different background concentrations of the Aitken mode particles (20 and 200 cm$^{-3}$). Each set contains five cases with different levels of accumulation mode particles (0, 3, 5, 10 and 20 cm$^{-3}$), i.e. 10 simulations in total. All particle concentrations are assumed to be constant with height. The simulations are named by a combination of two numbers, where the first number refers to the accumulation mode and the second to the Aitken mode number concentration (e.g., the case with 0 cm$^{-3}$ of accumulation and 20 cm$^{-3}$ of Aitken mode particles is referred to as the AC0_AK20 simulation). The concentrations are chosen to cover a range typical for aerosol size distributions in the summertime high Arctic (Heintzenberg and Leck, 2012; Leck and Svensson, 2015). If we assume that the accumulation mode contributes most to atmospheric CCN, then the simulation with 20 cm$^{-3}$ of accumulation mode aerosols agrees the best with the observations in terms of the CCN concentrations measured onboard the ship (Sect. 2.2). In this study, the simulation AC20_AK20 is defined as the baseline simulation.

A full description of meteorological conditions during the ASCOS campaign is available in Tjernström et al. (2012). During the ASCOS ice-drift period, observations were done over a surface dominated by pack ice and the surface conditions are thus set to represent sea ice in both models. The values used for the surface temperature and surface pressure are 269.8 K and 1026.3 hPa, respectively. The observed turbulent fluxes were usually smaller than 5 W m$^{-2}$ with peaks in the probability distributions around zero. Accordingly, the prescribed sensible and latent heat fluxes are set to zero (cf. also Stevens et al., 2018). The surface roughness is set to 0.0004 and the surface albedo to 0.844 (cf. Sedlar et al., 2011). In both models, the large-scale divergence is set to $1.5 \times 10^{-6}\,\mathrm{s}^{-1}$ at all model levels, which is the value required to obtain a stable cloud layer (cf. Stevens et al., 2018). There is no large-scale advection in the models. The aerosol population in both modes is represented by lognormal functions with the distribution parameters based on the ASCOS campaign measurements (Igel et al., 2017). Modal diameters of 32 and 93 nm and standard deviations and 1.1 and 1.5 are used for the Aitken and accumulation modes, respectively. The simulations are initialized with prescribed cloud droplet number concentration equal to 30 cm$^{-3}$ and the cloud water mixing ratios derived from the observations, i.e. a cloud is present at the beginning of all simulations. All microphysical processes are active at the beginning of the simulations in both models. The prescribed ice crystal concentration is 0.2 L$^{-1}$ (cf. Stevens et al., 2018). The model domain is three-dimensional and covers a region of $6 \times 6 \times 1.7$ km$^3$. In the horizontal direction, there is a fixed grid distance of 62.5 m in both models. In the vertical direction, the grid in MIMICA is variable with the highest resolution (7.5 m) at the surface and in the cloud layer. In RAMS, the vertical grid spacing is kept constant at 10 m. The different default configurations of vertical grid spacing in the two models could potentially generate discrepancies in the simulated cloud properties. However, we performed an additional baseline simulation with a fixed vertical resolution of 10 m with MIMICA and found no significant differences in the simulated results compared to the default version. The simulation period is 12 hours. The first 2 h are assumed to be a spin-up period and is therefore excluded from the figures and analysis. After the spin-up period, the cloud layer is stable in the baseline simulations.

In clean environments, the source of Aitken mode aerosols is typically new particle formation (NPF) and subsequent growth. In the high Arctic, however, different sources of Aitken mode aerosols have been proposed. Some studies associate the Aitken mode with the NPF events and subsequent growth by DMS oxidation products (e.g., Wiedensohler et al., 1996). Other studies suggest that the Aitken mode particles in this region are made up of marine gels produced by phytoplankton and sea-ice algae at the sea-surface interface (Leck and Bigg, 2005b). Different mechanisms imply different chemical compositions and thereby different hygroscopic properties that aerosol particles might have in the high Arctic. To study the impact of aerosol hygroscopicity, we performed additional simulations with different values of the hygroscopicity parameter, kappa, $\kappa$ (Petters and Kreidenweis, 2007). The default $\kappa$-value used to describe the hygroscopic properties of both aerosol modes is set to 0.4 (Leck and Svensson, 2015). As some previous studies (e.g., Christiansen et al., 2020) have shown that a change in hygroscopicity of the accumulation mode aerosols has almost no influence on the cloud properties, we only examined the sensitivity of the $\kappa$-value of the Aitken mode particles. Simulations AC3_AK20 and AC3_AK200 were performed with two additional $\kappa$-values = [0.1, 1.1], which cover a typical range of hygroscopicity of compounds expected to be present in high Arctic Aitken mode particles (Leck and Svensson, 2015). The lower limit of the hygroscopicity parameter tested ($\kappa$=0.1) would be representative of e.g., many organic compounds (e.g., Leck and Svensson, 2015) while the upper limit prescribed ($\kappa$=1.1) would correspond to a water-soluble inorganic salt like ammonium sulfate (Petters and Kreidenweis, 2007).

We also examined how the influence of Aitken mode particles on cloud microphysical properties depends on the amount of ice present in the cloud. Additional versions of the simulations AC3_AK20 and AC3_AK200 were performed with prescribed values of the ice crystal concentrations set to 0 and 1 L$^{-1}$. These values were chosen somewhat arbitrarily, but should generally represent a range describing an ice-free and an ice-rich cloud in the high Arctic. Note that the RAMS microphysical scheme includes hail and the one in MIMICA does not. However, the riming process in RAMS is inefficient for the examined conditions and pure ice crystals dominate the simulated ice water budget. The hail contribution to total surface precipitation is also two orders of magnitude lower than the rain contribution in RAMS.

## 3 Simulation results

### 3.1 Baseline simulation: comparison of simulated cloud properties

We first compare our baseline simulations (AC20_AK20) with time series of observed LWP and IWP (Fig. 2).

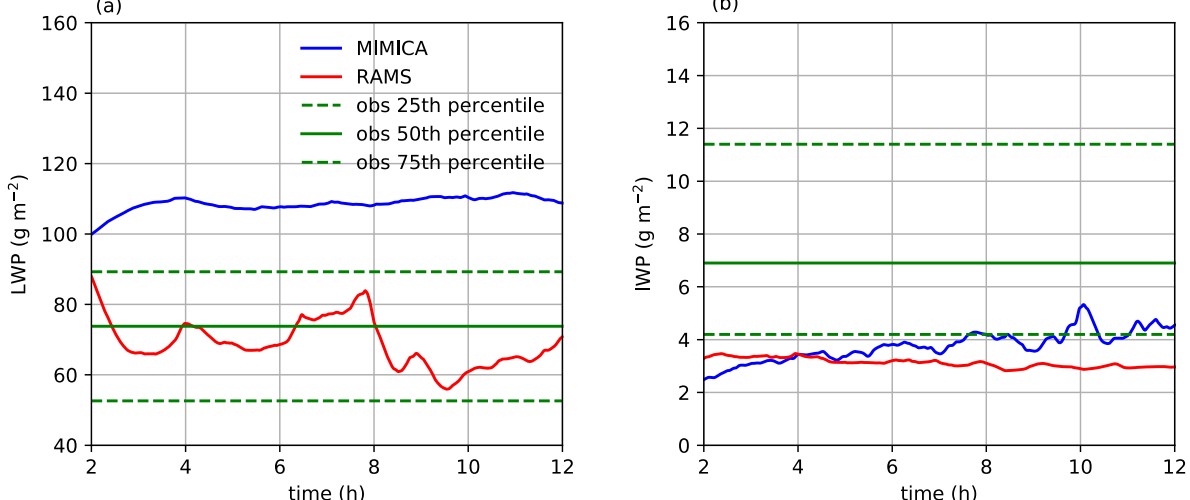

**Figure 2: (a) LWP and (b) IWP simulated by MIMICA and RAMS for the baseline simulations, i.e. with accumulation and Aitken mode concentrations of 20 cm⁻³. The retrieved values of LWP and IWP for the observed cloudy period defined in section 2.2 and up to the height of the model domains are shown as 25th, 50th (median) and 75th percentiles. The first 2 h of simulations, considered as spin-up, are excluded. The observed and simulated LWP and IWP include all liquid and ice water in the atmospheric column, i.e. water within and below the cloud.**

RAMS produces LWP values that fall within the observed range whereas MIMICA simulates a LWP that is 12–25 % higher than the 75th percentile of the observed range (Fig. 2a). In general, the use of prescribed aerosol particle concentrations should result in a higher LWP than if the simulations were performed with interactive aerosol particle concentrations (e.g., Stevens et al., 2018). It may be that MIMICA is more sensitive than RAMS in this regard. Furthermore, RAMS simulates weaker radiative cooling rates than MIMICA, which should produce a lower LWP in RAMS compared to MIMICA (see Sect. 3.2). The simulated IWP in both models is close to the 25th percentile of the observed range (Fig. 2b). In MIMICA, the IWP overlaps with the 25th percentile value in the second half of the simulation while in RAMS it is 17–33 % lower than the 25th percentile. Overall, the results show that both models simulate reasonable LWP and IWP compared to the observational data, however, it is hard to conclude which model performs better due to the large uncertainty and variability of the retrieved cloud variables (cf. also Sect. 2.2).

Simulated cloud droplet, rain and total ice mixing ratios for the baseline simulation are shown for the two models in Fig. 3. In both models, the cloud base (cloud top) height is the altitude above (below) which the cloud droplet mixing ratio exceeds the value $1\cdot10^{-6}$ kg/kg. The cloud droplet mixing ratio increases towards the top of the cloud in both models, but MIMICA produces slightly higher values in the upper part of the cloud layer (Fig. 3a and 3b), consistent with the higher LWP values (Fig. 2a). There is also a difference in the cloud top height evolution. In MIMICA, the cloud top height increases with time whereas it remains constant in RAMS. Rain mixing ratios are similar for the two models (Fig. 3c and 3d), but RAMS produces slightly more rain below the cloud. For the baseline simulations, the sum of the autoconversion rate and the collection rate of cloud droplets by rain drops is two orders of magnitude higher in RAMS than in MIMICA, which contributes to the higher rain mass mixing ratios in the RAMS sub-cloud layer. Both models also simulate similar values of total ice mixing ratios, although MIMICA produces a few stronger vertical bands after 6 h of simulation (Fig. 3e and 3f). This type of pronounced bands is a result of strong collection rates of rain drops by graupel, which appear at different times due to different temporal distributions of updrafts and downdrafts (Figure A8). To better understand the cloud dynamics, we have also examined the cloud diagnostics (i.e. mass transfer rates between gas and condensed phases) for cloud droplet water, rainwater and ice crystals (Fig. 4). In MIMICA, cloud droplet water has the highest condensation rates at the top of the cloud whereas in RAMS they are homogenously distributed within the cloud layer (Fig. 4a and 4b). The reason is most likely the higher entrainment rates at cloud top in MIMICA (not shown) that bring more water vapor into the cloud from the moist air that is present across the humidity inversion, which caps the cloud-topped boundary layer (Figure 1c). Higher entrainment rates are also consistent with the higher cloud top cooling rates present in MIMICA. Below cloud base, there is first a thin layer of cloud droplet evaporation in both models. In RAMS, there is also a sub-cloud condensation layer, which is produced by weak sub-cloud convection (not shown). Even though the condensation in this layer is infrequent, the associated mean rates are of the same order of magnitude as the condensation rates within the main cloud layer. The pockets of condensation and evaporation present in the main cloud layer are well-correlated with updrafts and downdrafts and they tend to cancel each other in the mean (not shown). This is why the average condensation rate in the main cloud is of the same order of

magnitude as the one in the sub-cloud layer in RAMS. However, if we consider the domain median instead of the mean, then the condensation rates are higher within the main cloud layer and they are zero below the evaporation layer also in RAMS (not shown). In both models, the condensational growth of raindrops is limited to the upper part of the cloud layer, while the maximum evaporation rates are found around the cloud base (Fig. 4c and 4d).

This typically happens when the environment is subsaturated for liquid water but supersaturated for ice, which results in evaporation of raindrops and growth of ice crystals. Ice crystals grow throughout the whole cloud layer with the highest deposition rates around cloud base (Fig. 4e and 4f), which corresponds well to the highest rain evaporation rates that are clearly seen in MIMICA (Fig. 4c). The ice crystal deposition and sublimation rates are higher in RAMS than in MIMICA since the two models partition the total ice deposition differently among ice

hydrometeor categories (not shown). Overall, the comparison shows that the simulated cloud microphysical properties are within the same order of magnitude for the two models, i.e. both models simulate the same cloud mechanisms leading to cloud dynamics that are similar in many aspects. However, there are still some notable differences that arise from the different model configurations.

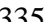

Figure 3: Cloud properties simulated by MIMICA and RAMS for the baseline simulation, i.e. with accumulation and Aitken mode concentrations of 20 cm⁻³. (a,b) cloud droplet mixing ratio (qc); (c,d) rain mixing ratio (qr); (e,f) total ice

(ice crystals, graupel (and hail for RAMS)) mixing ratio (qi total). The first 2 h of simulations, considered as spin-up, are excluded.. Black dashed lines represent the cloud top and cloud base heights.

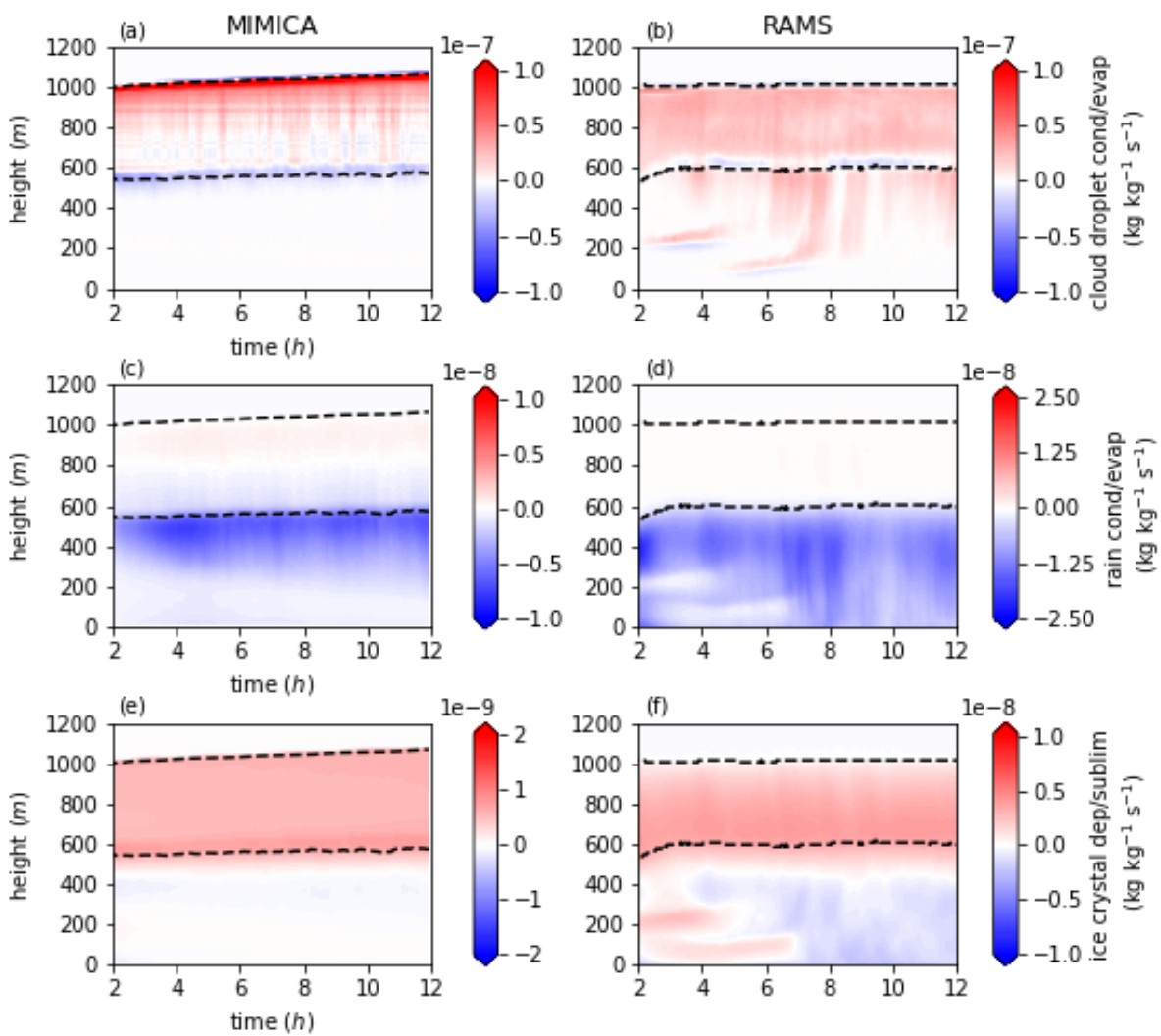

**Figure 4: Cloud diagnostics simulated by MIMICA and RAMS for the baseline simulation, i.e. with accumulation and Aitken mode concentrations of 20 cm[-3]. (a,b) cloud droplet condensation/evaporation rates; (c,d) rain drop condensation/evaporation rates; (e,f) ice crystal deposition/sublimation rates. The first 2 h of simulations, considered as spin-up, are excluded. Red color indicates net condensation/deposition and blue net evaporation/sublimation. Black dashed lines represent the cloud top and cloud base heights.**

### 3.2 Processes maintaining the simulated high Arctic SMP cloud

The cloud droplet mixing ratio for all simulations is shown in Fig. 5. In both models, the cloud thickens and the cloud base altitude change less with time when the number of accumulation mode aerosols increases. In general, MIMICA simulates a thicker cloud than RAMS. In both models, the cloud base altitude changes at the beginning of the simulation and it is especially pronounced in the cases with low accumulation mode particle concentrations.

However, the turbulence (Figure A6) is strong enough in MIMICA to develop and maintain a stable cloud during the whole simulation for all cases. This is not the case for RAMS where the cloud dissipates in the simulations with 0 cm$^{-3}$ of accumulation mode particles (i.e. the AC0_AK20 and AC0_AK200 simulations). As one of the main generators of cloud turbulence is radiative cooling at the top of the cloud, we have compared the cooling rates between the two models. They are about two to three times greater in MIMICA than in RAMS (Fig. 6 and

A2) and are more similar in MIMICA to values obtained from radiative transfer calculations based on observational data (Brooks et al., 2017). For the baseline case, both models simulate a relatively thick cloud (LWP > 40 g m$^{-2}$), which indicates that the differences in the cloud top cooling rates between MIMICA and RAMS do not arise from the difference in simulated LWP (cf., Garrett and Zhao, 2006). Moreover, the cooling rates in RAMS are smaller already at the beginning of the simulations when the liquid water contents are very similar in the two

models. A plausible explanation could instead be a less efficient radiative cooling parameterization in RAMS than in MIMICA. To further investigate the influence of the radiation parametrization on the model results, we performed additional simulations with simplified radiative transfer schemes in both MIMICA and RAMS as well as simulation with a prescribed higher cloud top cooling rate in RAMS (Appendix A, Fig. A1). These simulations show that the radiation parametrization significantly modifies the simulated liquid water content and could be the

cause of the observed differences between the models.

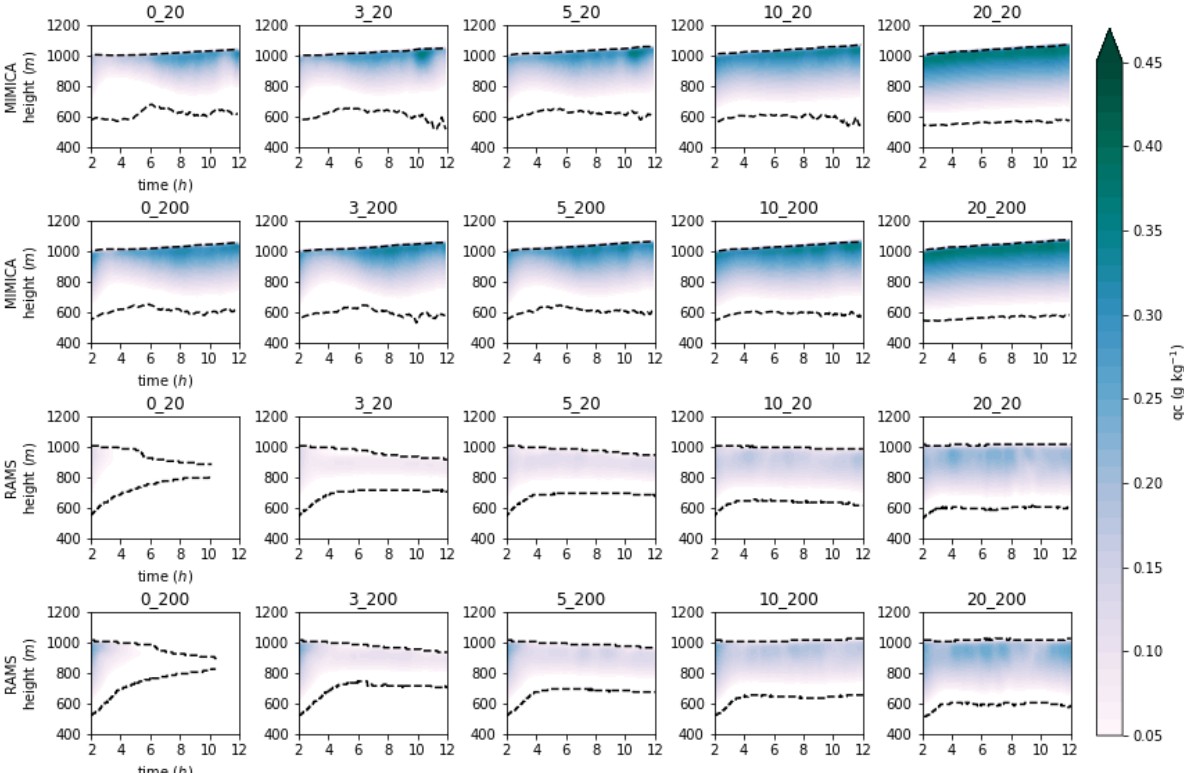

**Figure 5: Cloud droplet mixing ratio (qc) for the MIMICA and RAMS simulation sets. The first 2 h of simulations, considered as spin-up, are excluded. For figure clarity, the plot titles have been abbreviated; the first number refers to the accumulation mode and the second to the Aitken mode concentration in cm$^{-3}$, i.e. "0_20" refers to "AC0_AK20". Black dashed lines represent the cloud top and cloud base heights.**

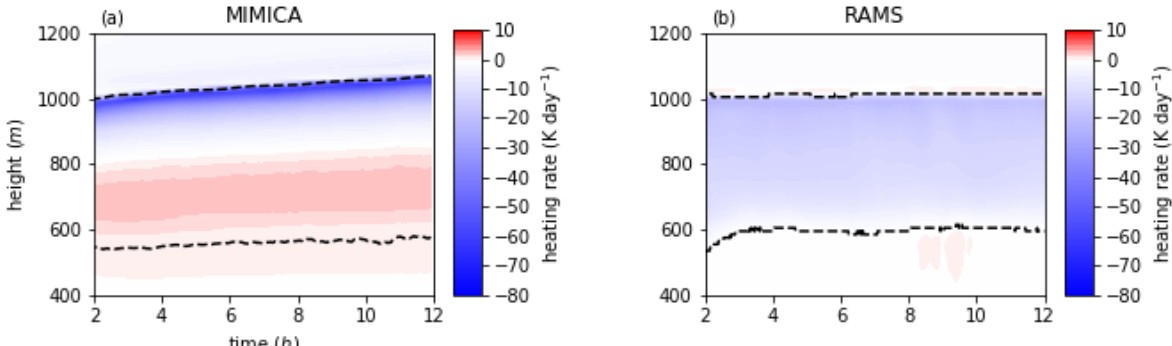

**Figure 6. Radiative heating rates in the baseline simulation (AC20_AK20), simulated by (a) MIMICA and (b) RAMS. The first 2 h of simulations, considered as spin-up, are excluded. Black dashed lines represent the cloud top and cloud base heights.**

### 3.3 Influence of Aitken mode particles

### 3.3.1 Influence of Aitken mode aerosol number concentration on cloud microphysical properties

To clearly show the influence of the Aitken mode particles on simulated cloud microphysical properties we plot the differences in cloud droplet (Fig. 7), rain (Fig. 8) and total ice (Fig.9) mixing ratios between each pair of simulations with the same accumulation mode concentration. The mass (Fig. 7) and number (not shown) of cloud droplet water generally increase in both models when Aitken mode particles are added, i.e. the particles serve as CCN and allow formation of additional cloud droplets. Aitken mode particles can also sustain the cloud at least for 6h when no accumulation mode particles are present (Fig. 7). The extent of their influence depends on the concentration of accumulation mode particles since these particles activate more easily and have the primary control on the cloud droplet number concentration. A higher number of cloud droplets decreases the maximum supersaturation and the amount of water vapor in the cloud available for activation of smaller particles. In both models the influence of smaller particles on cloud droplet mixing ratio thus generally decreases with increasing accumulation mode concentration (Fig. 7). Nevertheless, the differences in cloud droplet mixing ratio are statistically significant for all pairs of different Aitken mode concentrations in both models, except for the MIMICA pair AC20_AK200 and AC20_AK20 (according to a student t-test with a 95 % confidence level on the time averages in the cloud layer). In other words, both models show that Aitken mode particles have a significant impact on the cloud droplet mixing ratio, at least up to 20 cm$^{-3}$ of accumulation mode particles in RAMS and at least up to 10 cm$^{-3}$ in MIMICA. At the cloud top, a distinct maximum difference occurs in both models as a result of higher cloud top heights in the simulations with a higher Aitken mode concentration (Fig. A3).

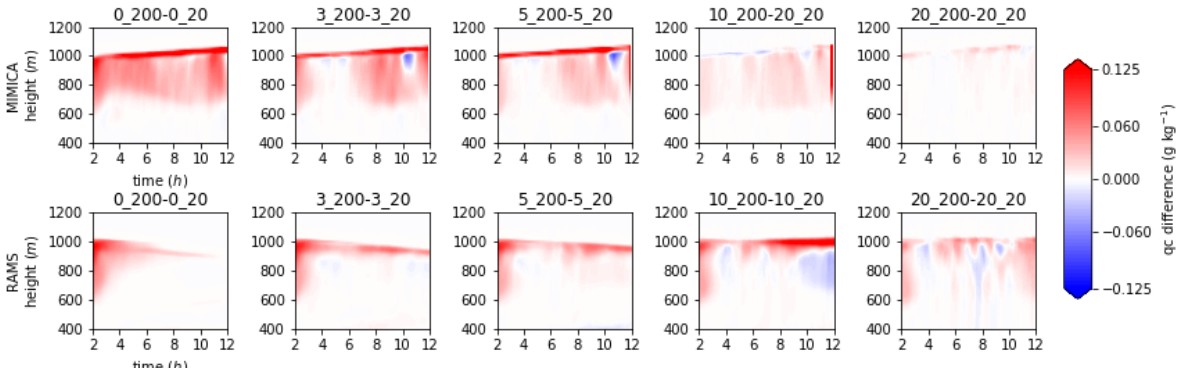

**Figure 7: Differences in cloud droplet mixing ratio (qc) for simulation pairs with the same accumulation mode concentration (i.e. ACx_AK200-ACx_AK20) shown for MIMICA and RAMS. The first 2 h of simulations, considered as spin-up, are excluded. A student's t-test with a 95% confidence level shows that the (time mean) differences are statistically significantly different for each pair of simulations, except for the pair AC20_AK200 and AC20_AK20 in MIMICA. For figure clarity, the plot titles have been abbreviated; the first number refers to the accumulation mode and the second to the Aitken mode concentration in cm$^{-3}$, i.e. "0_20" refers to "AC0_AK20".**

Most of the rain water is present in the upper part of the cloud layer in all simulations with MIMICA (Fig. A4), in line with the maximum rain condensation rates shown for the baseline simulation (Fig. 4). In RAMS, the cases with higher accumulation mode concentrations, i.e. with a stable cloud, also show that most of the rain water is present close to the cloud top (Fig. A4). Both models produce both positive and negative differences in rain water mixing ratio that vary with time with increasing Aitken mode particle concentrations (Fig. 8). At the beginning of
the simulations, there is in general more rain produced in the cases with a higher number of Aitken mode particles (i.e. positive differences). An increase in the Aitken mode particle concentration leads to stronger turbulence (Fig. A6) and more cloud liquid water production (Fig. 5), which leads to stronger rain rates at the beginning of the simulations (cf. also Fig. A4). Towards the end of the simulations, the rain rates are either about the same or there is less rain (i.e. negative differences). The presence of both positive and negative differences with time is a result
of different temporal evolutions of updrafts and downdrafts in each individual simulation, which influences the production of rain and ice in the cloud (cf. Fig. A8b,c).

       The total ice mixing ratios for all simulations are presented in Fig. A5, while the differences due to different Aitken mode concentrations are shown in Fig. 9. For the two lowest accumulation mode concentrations (0 and 3 cm$^{-3}$), there is in general more ice with a higher number of Aitken particles in both models. In the cases with more
accumulation mode particles (e.g., 5, 10, 20 cm$^{-3}$), the variability is larger with both positive and negative differences. This result can be related to differences in cloud dynamics; the maximum updrafts are reached at somewhat different times in the two models (cf. Figure A8b,c). The influence of Aitken mode particles on ice is in general larger in MIMICA than in RAMS consistent with the stronger cloud top cooling rates simulated by MIMICA that favors the ice formation through immersion freezing and growth by vapor deposition when the
number of CCN increases (Possner et al. 2017; Solomon et al., 2018; Eirund et al., 2019).

       The shown influence of Aitken mode particles on cloud microphysical properties and cloud sustenance has implications for the surface energy fluxes (Fig. 10). The influence of the smaller particles on the LW fluxes decreases as the number of accumulation mode particles increases (i.e. smaller differences) in both models, but it is statistically significant in all cases except for the pair AC20_AK200 and AC20_AK20 in MIMICA. The results
are consistent with the influence of Aitken mode on cloud droplet mixing ratios (cf. Fig. 7). Both models simulate no significant influence of Aitken mode particles on the SW radiation, consistent with the low insolation (not shown).

       We also tested the sensitivity of the simulated cloud properties to different Aitken mode particle concentrations for different levels of ice crystal concentrations (see Sect. 2.3). These simulations show that the influence of the
Aitken mode particles on the liquid phase decreases in clouds with more ice (Fig. 11 and 12). This result agrees well with previous studies that have investigated the influence of CCN in mixed-phase clouds with different background INP or ice crystal concentrations (e.g., Possner et al., 2017; Stevens et al., 2018).


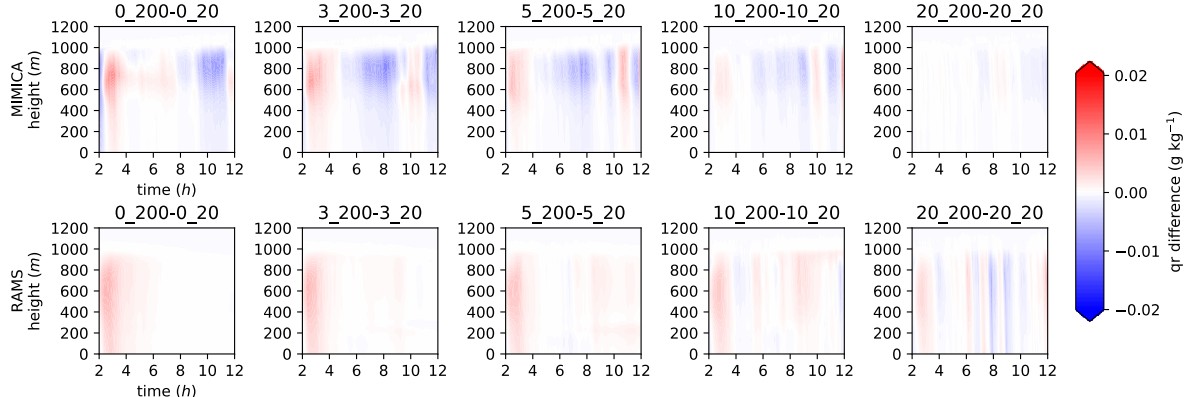

**Figure 8: Differences in rain mixing ratio (qr) for simulation pairs with the same accumulation mode concentration (i.e. ACx_AK200-ACx_AK20) shown for MIMICA and RAMS. The first 2 h of simulations, considered as spin-up, are**
**excluded. A student's t-test with a 95% confidence level shows that the (time mean) differences are statistically significantly different for each pair of simulations except for the pair AC20_AK200 and AC20_AK20 in MIMICA. For figure clarity, the plot titles have been abbreviated; the first number refers to the accumulation mode and the second to the Aitken mode concentration in cm$^{-3}$, i.e. "0_20" refers to "AC0_AK20".**


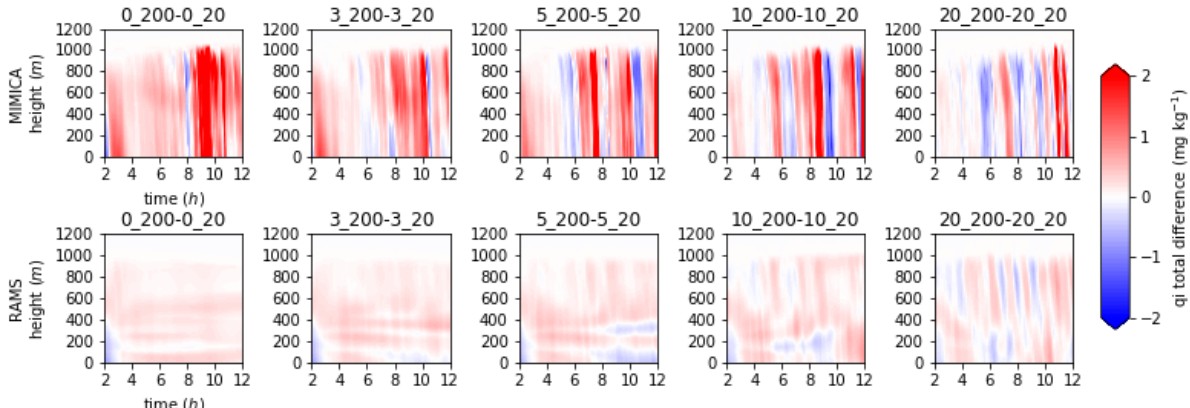

**Figure 9: Differences in total ice mixing ratio (qi total) for simulation pairs with the same accumulation mode concentration (i.e. ACx_AK200-ACx_AK20) shown for MIMICA and RAMS. The first 2 h of simulations, considered as spin-up, are excluded. A student's t-test with a 95% confidence level shows that the (time mean) differences are statistically significantly different in the first four pairs of simulations in both models. For figure clarity, the plot titles have been abbreviated; the first number refers to the accumulation mode and the second to the Aitken mode concentration in cm$^{-3}$, i.e. "0_20" refers to "AC0_AK20".**

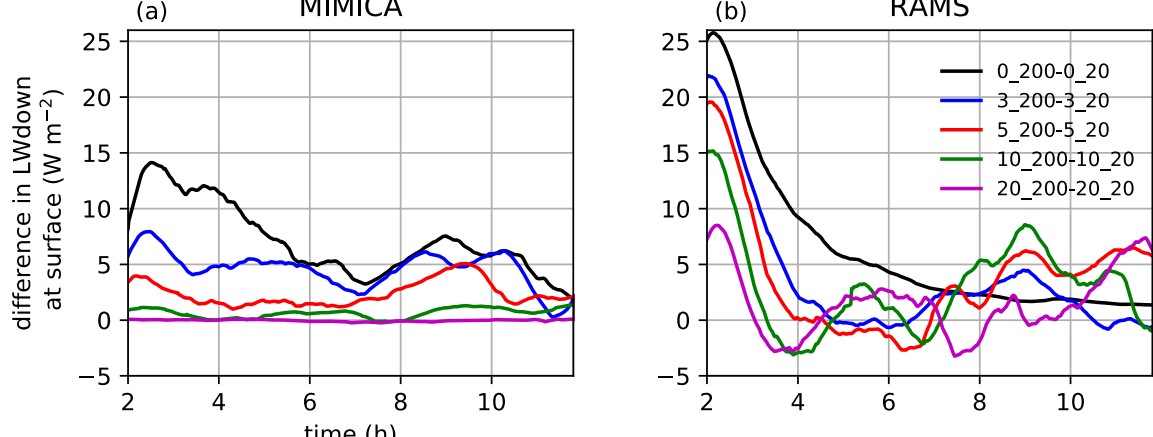

**Figure10: Difference in downward longwave (LWdown) radiation at the surface for simulation pairs with the same accumulation mode concentration (i.e. ACx_AK200-ACx_AK20) shown for (a) MIMICA and (b) RAMS. The first 2 h of simulations, considered as spin-up, are excluded. A student's t-test with a 95% confidence level shows that the (time mean) differences are statistically significantly different for each pair of simulations except for the pair AC20_AK200 and AC20_AK20 in MIMICA. For figure clarity, the plot titles have been abbreviated; the first number refers to the accumulation mode and the second to the Aitken mode concentration in cm$^{-3}$, i.e. "0_20" refers to "AC0_AK20".**

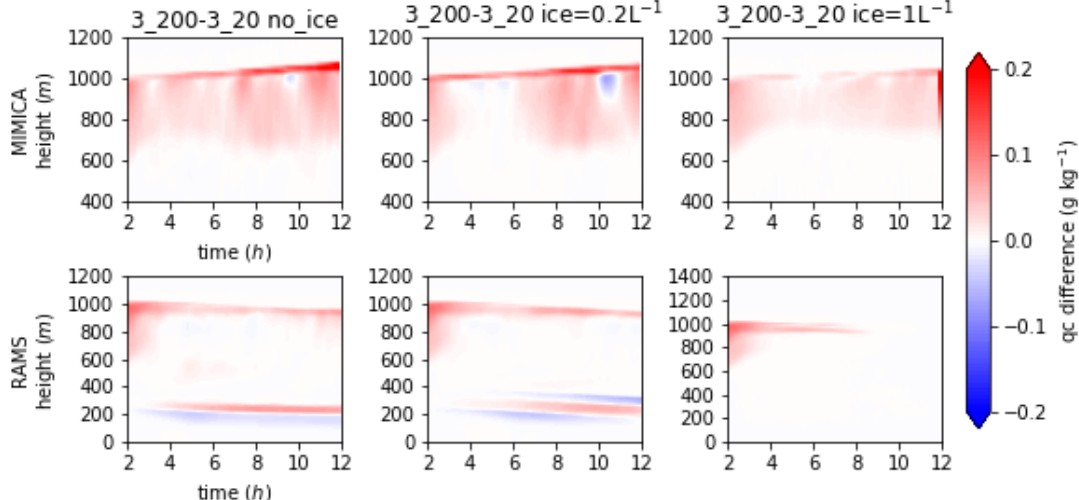


**Figure 11: Differences in cloud water mixing ratio (qc) for simulation pairs with the same accumulation mode concentration and the same ice crystal concentration: =0 L⁻¹ (no_ice, the leftmost column); =0.2 L⁻¹ (the middle column); =1 L⁻¹ (the rightmost column) shown for MIMICA and RAMS. The first 2 h of simulations, considered as spin-up, are excluded. A student's t-test with a 95% confidence level shows that the (time mean) differences are statistically significantly different for each pair of simulations. For figure clarity, the plot titles have been abbreviated; the first number refers to the accumulation mode and the second to the Aitken mode concentration in cm⁻³, i.e. "3_20" refers to "AC3_AK20".**



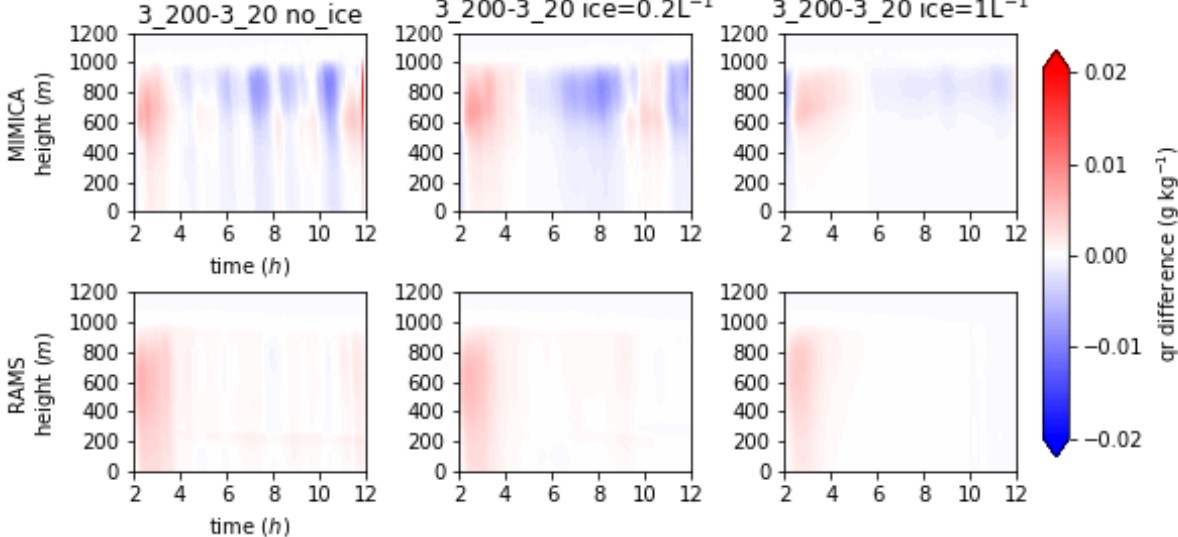


**Figure 12: Differences in rain water mixing ratio (qr) for simulation pairs with the same accumulation mode concentration and the same ice crystal concentration: =0 L⁻¹ (no_ice, the leftmost column); =0.2 L⁻¹ (the middle column); =1 L⁻¹ (the rightmost column) shown for MIMICA and RAMS. The first 2 h of simulations, considered as spin-up, are excluded. A student's t-test with a 95% confidence level shows that the (time mean) differences are statistically significantly different for each pair of simulations except for the pair with ice crystal concentration of 1 L⁻¹ in RAMS. For figure clarity, the plot titles have been abbreviated; the first number refers to the accumulation mode and the second to the Aitken mode concentration in cm⁻³, i.e. "3_20" refers to "AC3_AK20".**



**3.3.2 Influence of Aitken mode aerosol hygroscopicity on cloud microphysical properties**

Figure 13 shows that the change in the cloud droplet mixing ratio induced by Aitken mode particles increases as their κ-value increases, i.e. more hygroscopic Aitken mode particles lead to a larger increase in the cloud droplet mass. The cloud droplet number undergoes the same dependence (not shown). Higher particle hygroscopicity allows aerosol particle activation at lower supersaturations (see Sect. 4 for more information on supersaturation statistics).

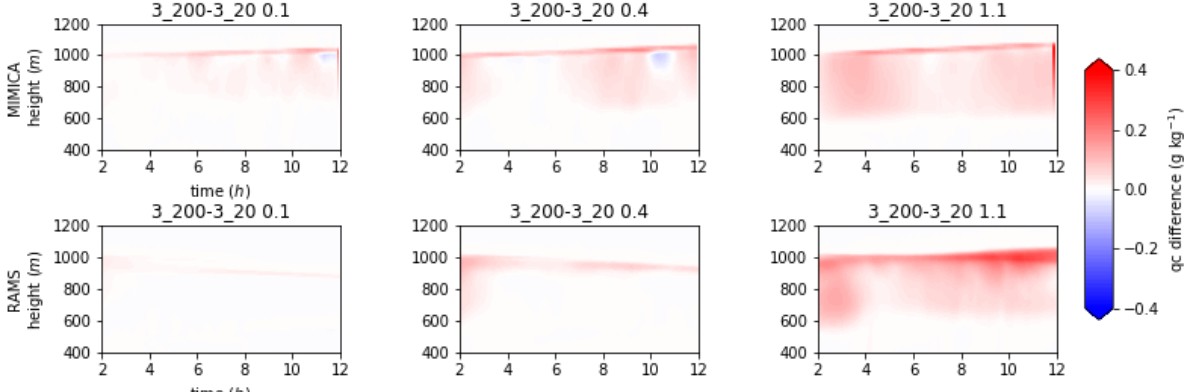

**Figure 13: Differences in cloud droplet mixing ratio (qc) for simulation pairs with the same accumulation mode concentration and the same kappa value of the Aitken mode particles: =0.1 (the leftmost column); =0.4 (the middle column); =1.1 (the rightmost column) shown for MIMICA and RAMS. The first 2 h of simulations, considered as spin-up, are excluded . A student's t-test with a 95% confidence level shows that the (time mean) differences are statistically significantly different for each pair of simulations. For figure clarity, the plot titles have been abbreviated; the first number refers to the accumulation mode and the second to the Aitken mode concentration in cm$^{-3}$, i.e. "0_20" refers to "AC0_AK20".**

The addition of Aitken mode particles with high κ-value leads to negative differences in rain amount in MIMICA, which can be explained by a greater number of cloud droplets and less efficient production of rain drops. However, in RAMS the differences are mostly positive, i.e. there is an increase in rain water mixing ratio (Fig. A9). The reason for this is most likely the very weak cloud layer produced by RAMS in the original AC3_AK20 simulation. As there is no cloud, there is also almost no precipitation - regardless of the κ-value. In both models, the impact of Aitken mode particles on the total ice mixing ratio generally becomes greater as the hygroscopicity of the particles increases (Fig. A10).

To summarize, the sensitivity tests show that Aitken mode particles can be activated even with a κ-value equal to 0.1 (more pronounced in MIMICA). Based on the model simulations, we can thus conclude that Aitken mode particles do not have to be highly hygroscopic in order to become CCN-active if accumulation mode aerosol concentrations are low.

**4 Supersaturation statistics**

We analyze next the simulated water vapor supersaturation (ss) values reached within the model domains in order to investigate how the ss statistics depend on different prescribed aerosol size distributions as well as on different hygroscopic properties of the Aitken mode particles. The ss statistics are calculated for a 20min period around 6 h of simulation for all the cases simulated with the default κ-value (κ=0.4; i.e. dependence on the aerosol size distribution, Fig. 14). They are also calculated for the AC3_AK20 and AC3_AK200 simulations initialized with different κ-values for the Aitken mode particles (κ=[0.1,1.1]; i.e. dependence on the hygroscopic properties, Fig. 15). In figure 14, the median ss values in both models generally vary between 0.2 and 0.4 %. The exception is the case AC0_AK20 in RAMS where there is no stable cloud at 6 h. The ss values in this simulation are high since the statistics are based on a relatively low number of supersaturated grid boxes, which in this case reach high ss values due to a low condensational sink. The median numbers agree well with typical ss values reported for clean marine stratocumulus clouds at mid-latitudes (Hudson and Noble, 2014; Yang et al., 2019). However, the 99[th] percentiles show high supersaturations with values above 1 % for most of the simulations. As expected, simulated ss values decrease with higher accumulation mode number concentration. They are even lower when the Aitken

mode concentration is prescribed to a larger number (200 vs. 20 cm$^{-3}$). The median values in Fig. 15 also vary between 0.2 and 0.4 % and in general decrease with a higher κ-value of the Aitken mode particles for the two tested concentrations in both MIMICA and RAMS. Again, the 99$^{th}$ percentiles show high values that exceed 1 % in most of the cases.

The numbers shown in Fig. 14 and Fig. 15 are the critical dry diameters calculated for the 75$^{th}$ and 99$^{th}$ percentiles of the ss values (cf. Fig. A10). These values can be compared with the mean diameter of 32 nm prescribed for the Aitken mode in this study. Our analysis confirms that supersaturations within the model domains reach high enough values to activate Aitken mode particles for all tested accumulation mode concentrations and all tested Aitken mode κ-values. If the calculations in Fig. A14 are done for higher (lower) surface tension, the maximum ss values would need to be higher (lower) to activate particles of the same critical dry diameters as the ones presented here.

Updraft statistics calculated for the same time period as the ss statistics for the set of cases simulated with the default κ-value show that the updrafts are in general stronger with increasing accumulation mode concentration (Fig. A11). The updraft values generally cover a range between 0 to 1 ms$^{-1}$, which agrees well with the vertically resolved updraft estimates by Sedlar and Shupe (2014) for the ASCOS campaign. With an increase in accumulation mode particles, there is more vapor condensation and thus more liquid water in the mixed-phase cloud, which drives the turbulence through cloud-top radiative cooling (cf. Possner et al., 2017; Stevens et al., 2018). Stronger turbulence further leads to stronger updrafts and further condensation.

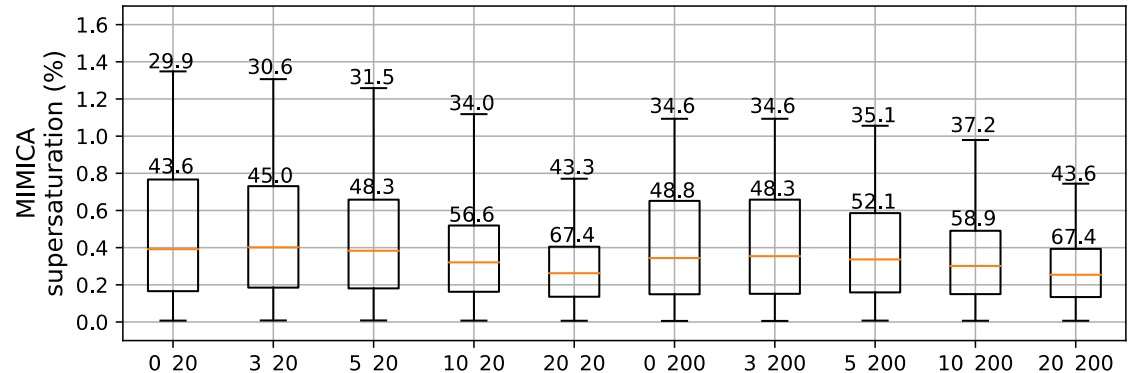

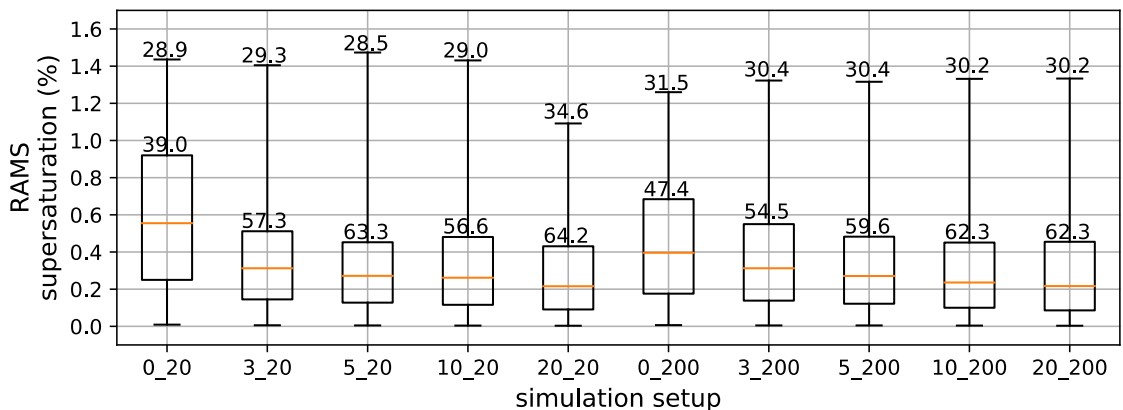

**Figure 14: Supersaturation statistics shown for a set of cases initialized with a κ-value of 0.4, simulated by MIMICA and RAMS. The statistics are calculated for a 20min period around 6 h of simulation for all grid boxes with relative humidity > 100 %. Lower and upper whiskers correspond to 1$^{st}$ and 99$^{th}$ percentiles, respectively. The numbers written in the figure are critical dry diameters that correspond to supersaturation 75$^{th}$ percentiles (upper limit of the box) and 99$^{th}$ percentiles. For figure clarity, the plot titles have been abbreviated; the first number refers to the accumulation mode and the second to the Aitken mode concentration in cm$^{-3}$, i.e. "0_20" refers to "AC0_AK20".**

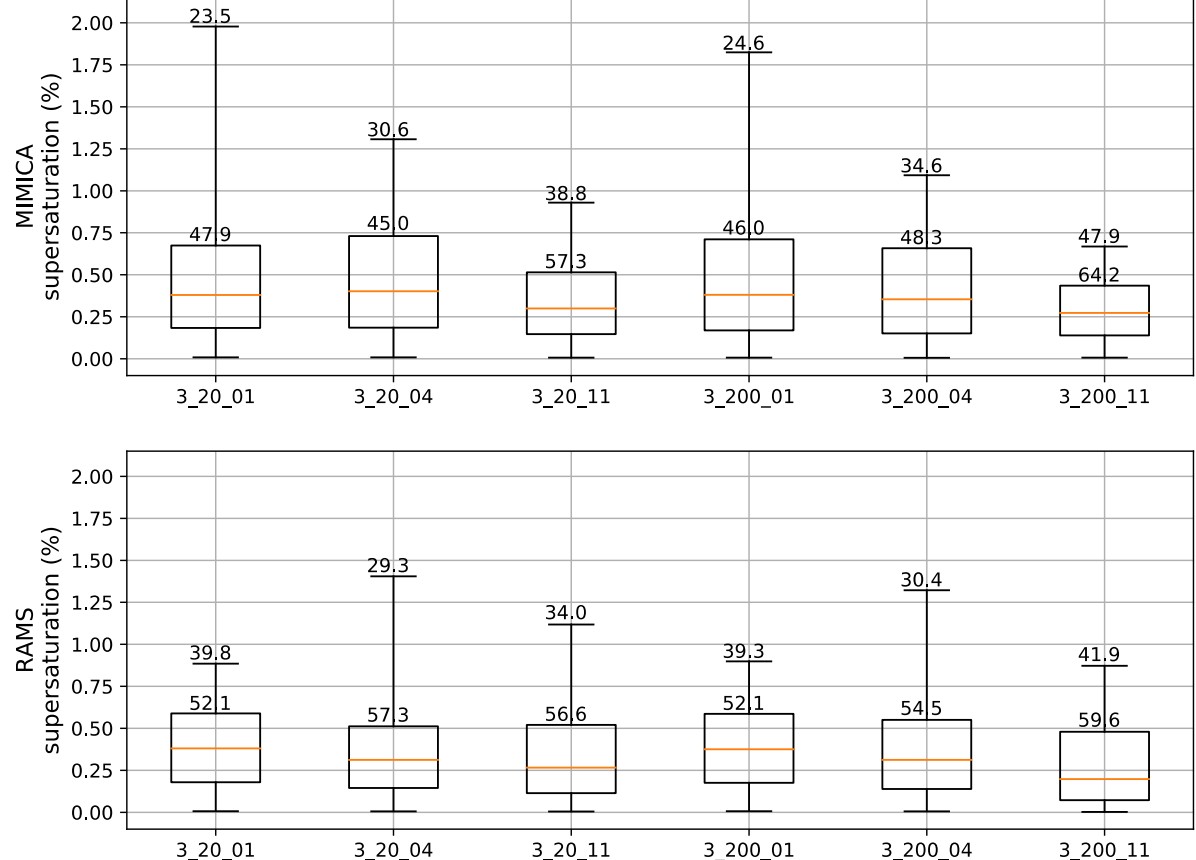

**Figure 15. Supersaturation statistics shown for the simulations AC3_AK20 and AC3_AK200 initialized with different κ-values = [0.1, 0.4, 1.1], simulated by MIMICA and RAMS. The statistics are calculated for a 20min period around 6 h of simulation for all grid boxes with relative humidity > 100 %. Lower and upper whiskers correspond to 1st and 99th percentiles, respectively. The numbers written in the figure are critical dry diameters that correspond to supersaturation 75th percentiles (upper limit of the box) and 99th percentiles. For figure clarity, the plot titles have been abbreviated; the first number refers to the accumulation mode and the second to the Aitken mode concentration in cm-3, i.e. "3_20_01" refers to "AC3_AK20" initialized with a κ-value=0.1.**

## 5 Discussion

### 5.1 Qualitative comparison of model results with observational data for the High Arctic

Both models suggest that Aitken mode particles are important as CCN in summertime high Arctic SMP clouds if accumulation mode concentrations are low. Guided by these analyses we have revisited the observed aerosol size distributions from four high Arctic expeditions, including the ASCOS campaign (Leck et al., 1996; Leck et al., 2001; Leck et al., 2004; Tjernstöm et al., 2014). We first examined the representativeness of the size distributions that we have applied in our simulations, i.e. how frequently these types of distributions occur in the observations. Fig. 16 shows two classes of size distributions: one with Aitken mode concentrations lower than 25 cm-3 (AIT < 25 cm-3, blue line) and one with Aitken mode concentrations between 100<AIT<200 cm-3 (orange line). The cases with accumulation mode number concentrations equal to 20 cm-3 (i.e. the maximum accumulation mode concentration prescribed in the simulations) have the occurrence probability of 5 % and 17 % (of total minutes of observations) for the class 100<AIT<200 cm-3 and AIT < 25 cm-3, respectively. This means that in conditions with low accumulation mode concentrations (i.e. lower than 20 cm-3) there is a higher probability for the Aitken mode particle concentration to also be low in number (i.e. lower than ~25 cm-3). However, it also happens that Aitken mode concentrations are much higher (>~100 cm-3). In other words, the prescribed size distributions that we have applied in our simulations are reasonable.

Probability density functions (PDFs) of observed Hoppel diameters (Hoppel et al., 1986; Fig. 17), calculated as detailed in Heintzenberg and Leck (2012), show that the PDFs for all four expeditions peak around 60 nm, i.e. this

should be the most common activation diameter. However, the smallest observed Hoppel diameters are around 40 nm, supporting our conclusions that small Aitken mode particles may be activated in the summertime high Arctic under certain conditions. The observational statistics agree well with the calculations of the critical dry diameters obtained from the simulated ss values (Sect. 4).

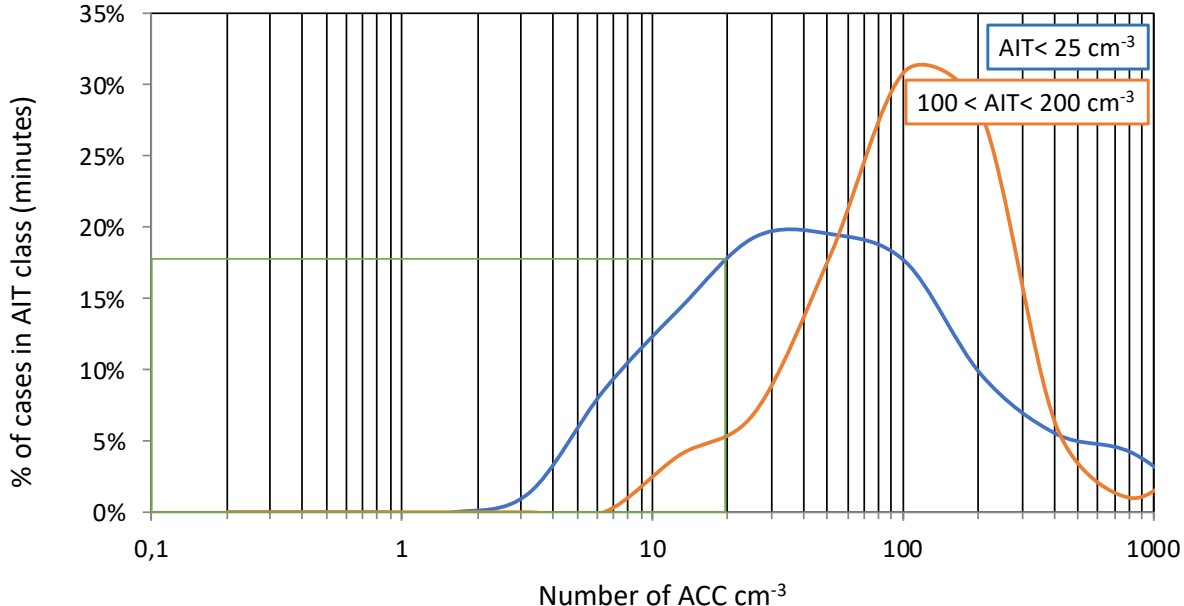

**Figure 16. The occurrence probability (% of total minutes of observations) for two classes of Aitken (AIT) mode concentrations: AIT < 25 cm⁻³ and 100<AIT<200 cm⁻³. On the x-axis is the number of accumulation (ACC) mode particles in cm⁻³. The statistics are calculated for four different expeditions in the high Arctic, in the summers of 1991,**
**1996, 2001 and 2008. Further details on the quality and data processing of the aerosol size resolved measurements are available in Heintzenberg and Leck (2012).**

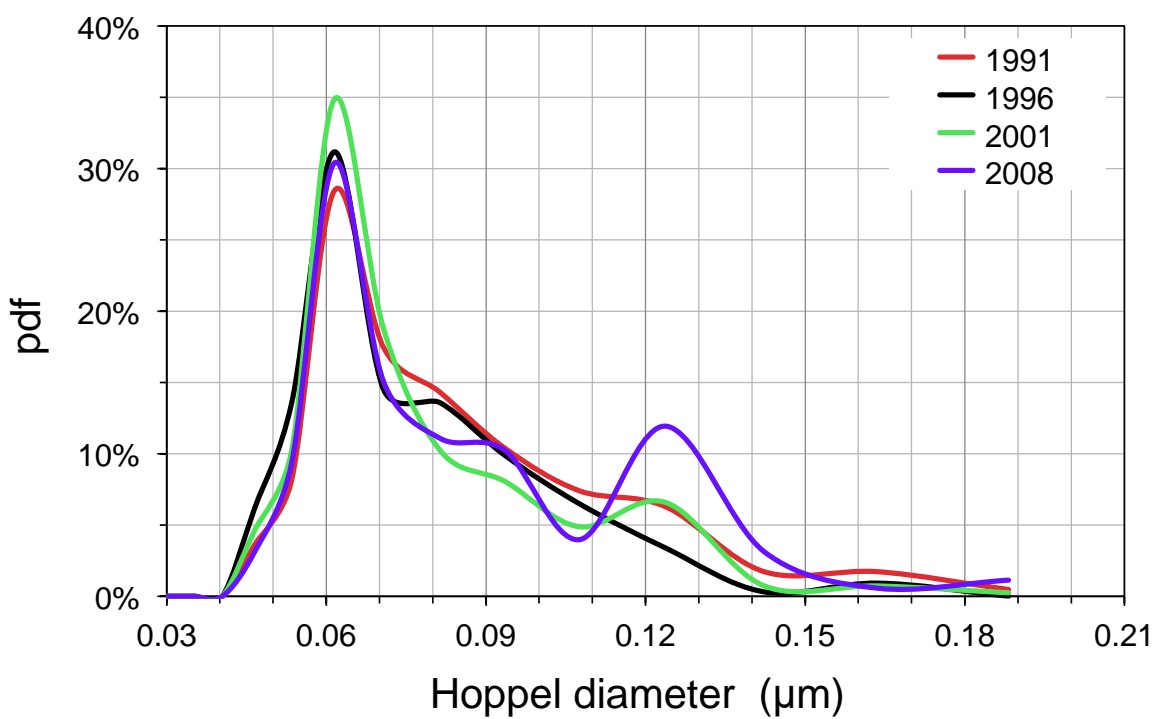

**Figure 17: Probability density function (pdf) of the Hoppel diameter shown for four different expeditions in the high Arctic, in the summers of 1991, 1996, 2001 and 2008. Further details on the quality and data processing of the aerosol size resolved measurements are available in Heintzenberg and Leck (2012).**

## 5.2. General importance of Aitken mode particles for low-level mixed-phase cloud properties

Our study focuses on the summertime, ice-covered, high (> 80° N) Arctic region. However, it is reasonable to assume that the results are also valid for low-level mixed-phase clouds in other regions with low (<10-20 cm$^{-3}$) accumulation mode aerosol concentrations. The activation diameters derived in Section 4 support recent findings for the region south of the ice edge during summertime, which show that particles smaller than the accumulation mode potentially can act as CCN (i.e. smaller than 50 nm in diameter; Willis et al., 2016; Kecorius et al., 2019; Koike at al., 2019). Several studies have investigated the seasonality of aerosol particle size distributions in the Arctic (e.g., Tunved et al., 2013; Freud et al., 2017, Koike at al., 2019). They show that number concentrations of accumulation mode particles are lowest during the summer and autumn months and that they can reach values below <10-20 cm$^{-3}$ at several locations in the Arctic. However extremely low accumulation mode number concentrations (occasionally below 1 cm$^{-3}$) have only been found in the high Arctic (Bigg et al., 1996; Mauritsen et al., 2011; Heintzenberg et al., 2015; Leck and Svensson, 2015). During summertime, conditions in the Arctic are generally favourable for NPF (Tunved et al., 2013; Croft et al., 2016; Nguyen et al., 2016) and local marine sources are active (Leck and Bigg, 2005a; Heintzenberg and Leck, 2012; Karl et al., 2013; Heintzenberg et al., 2015), supporting the presence of high concentrations of small particles. It is thus likely that Aitken mode particles are most important during the summer months (high concentrations of Aitken mode aerosols) and over remote areas covered by ice or snow (low accumulation mode aerosol concentrations).

Note that the simulated influence of Aitken mode particles can also be dependent on details in the simulation setup. In this study, there are no sources or sinks of aerosols during the simulation time, the aerosols are only passively advected within the model domains (cf. Sect. 2.3). If aerosol sinks were included, the influence of Aitken mode particles would most likely be even more pronounced since accumulation mode aerosols are more efficient as CCN and should be removed faster from the cloud than the Aitken mode particles. Furthermore, the dependence of the Aitken mode influence on the cloud ice amount is investigated here based on different, prescribed ice crystal concentrations (cf. Sect. 2.3). The results most likely depend on whether the ice crystal concentrations are prognostic or prescribed and and if secondary ice processes are considered in the calculations of the ice crystal number concentration (Sotiropoulou et al., 2020).

## 6 Summary and conclusions

This study investigates the potential importance of Aitken mode particles in sustaining and affecting the properties of stratiform mixed-phase clouds in the summertime high Arctic. To perform such a task, we have used two LES models (MIMICA and RAMS) to simulate a high Arctic SMP cloud observed during the ASCOS campaign (Tjernström et al., 2014) and initialized the models with different aerosol size distributions. Both models show that Aitken mode aerosols have a significant impact on the simulated cloud droplet mixing ratio, if the accumulation mode number concentration is less than 10–20 cm$^{-3}$. Simulations performed with different values of the hygroscopicity parameter κ indicate that more hygroscopic Aitken mode particles lead to a higher amount of cloud droplet water, as expected. Moreover, the simulations show that Aitken mode particles can act as CCN and influence the properties of SMP clouds even at the low κ-values (=0.1). If the ice fraction of the SMP cloud is high (i.e. ice-rich clouds), the influence of Aitken mode particles on the liquid phase decreases, corroborating the results by Possner et al. (2017) and Stevens et al. (2018).

Both models are in qualitative agreement in terms of the influence of Aitken mode particles on cloud properties, even though the models show different results regarding e.g., the simulated amount of liquid water and the relative role of different microphysical processes governing the overall cloud properties. The most striking difference between the two models appears to be caused by a difference in the radiation schemes. RAMS produces less radiative cooling for a certain amount of cloud water compared to MIMICA and does not sustain a cloud at low accumulation mode aerosol concentrations (<3–10 cm$^{-3}$). The radiative cooling rates produced by MIMICA agree better with the observation-based estimates by Brooks et al. (2017), but the observations are in general not sufficient to constrain or rank the models in terms of their performance. This would require additional observations (of e.g., cloud-top radiative cooling rates, updrafts, supersaturation values) and less uncertainty in the retrieved data (of e.g., LWP and IWP).

The simulated median supersaturations in both MIMICA and RAMS vary between 0.2 and 0.4 %, but values above 1 % were also found within the model domains (99[th] percentile values). The spatial variability in the simulated supersaturations and updrafts demonstrates the potential issue of applying constant supersaturation values for a grid box, or even a certain cloud type, within e.g., general circulation models. Calculations of threshold

diameters of aerosol activation confirm that the simulated supersaturation values are high enough for Aitken mode particles to be activated (i.e. the activation diameter is as low as ~30 nm). Furthermore, statistics of the observed Hoppel minimum diameter from four different expeditions in the high Arctic (Heintzenberg and Leck, 2012) also suggest that aerosols in the Aitken mode are activated as CCN. Our results are in qualitative agreement with recent studies for the lower Arctic, which indicate that particles smaller than 50 nm act as CCN (Willis et al., 2016;
Kecorius et al., 2019; Koike at al., 2019), and thus suggest that Aitken mode aerosols more generally influence mixed-phase cloud properties in environments with low accumulation mode aerosol concentration.

Our findings highlight the importance of better understanding Aitken mode particle formation, chemical composition and emissions, in particular in pristine environments such as the high Arctic in summer. It is reasonable to assume that the influence of these particles can be significant in any environment and during other
seasons when the accumulation mode particle concentrations are low. The results show that accumulation mode particles should not be considered as the only potential CCN in models, as this may lead to e.g., too low background CCN concentrations and too high estimates of anthropogenic aerosol indirect effects.










**Appendix A:**

Cloud droplet mixing ratios simulated by the two models using different radiative transfer schemes is shown in Fig. A1. Using simple radiative transfer schemes (i.e. radiation_simple simulations; the radiative fluxes depend
on LWP only, Stevens et al. (2005) in MIMICA and Chen and Cotton (1983) in RAMS) instead of the default radiation solvers (radiation_solver simulations; Fu and Liou (1993) in MIMICA and Harrington (1997) in RAMS) leads to a lower cloud water amount and a thinner cloud in MIMICA compared to RAMS, i.e. the opposite result compared to when using the default radiation solvers. Another test where the radiative cooling rates within RAMS were multiplied by a factor of 5 at the top of the cloud produces a much thicker cloud than the one in the MIMICA
radiation_solver simulation, which confirms that the cooling efficiency of the radiative scheme is a critical factor for determining the cloud droplet amount and consequently also the cloud lifetime. The results show that the radiation parametrization used in the model has a significant impact on the simulated cloud properties and is especially important to be considered in model intercomparison studies.


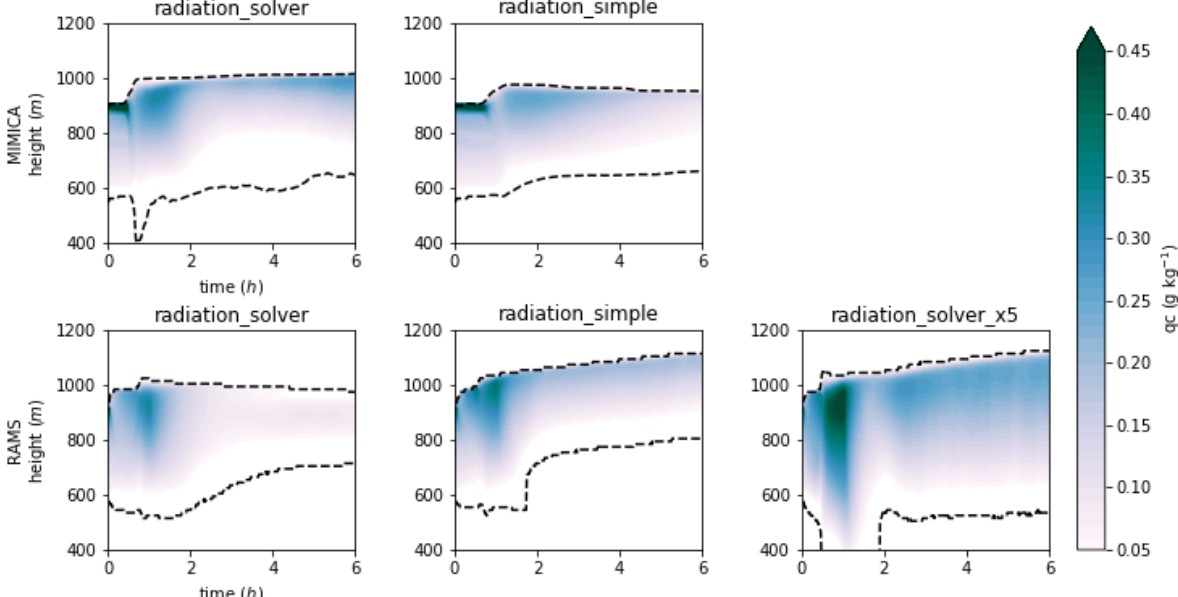

**Figure A1: Cloud droplet mixing ratio (qc) shown for a simulation AC3_AK20 initialized with different radiative schemes in MIMICA and RAMS. The title *radiation_solver* is used for the simulations where the models are initialized with their default radiation solvers (Fu and Liou (1993) in MIMICA and Harrington (1997) in RAMS). The title *radiation_simple* is used for the simulations where the radiative fluxes are calculated as functions of LWP only (Stevens et al. (2005) in MIMICA and Chen and Cotton (1983) in RAMS). The *radiation_solver_x5* simulated by RAMS shows**
**the qc obtained with the default radiation solver but with a 5x higher cooling rate enforced at cloud top. The simulations are run for 6 h. Black dashed lines represent the cloud top and cloud base heights.**


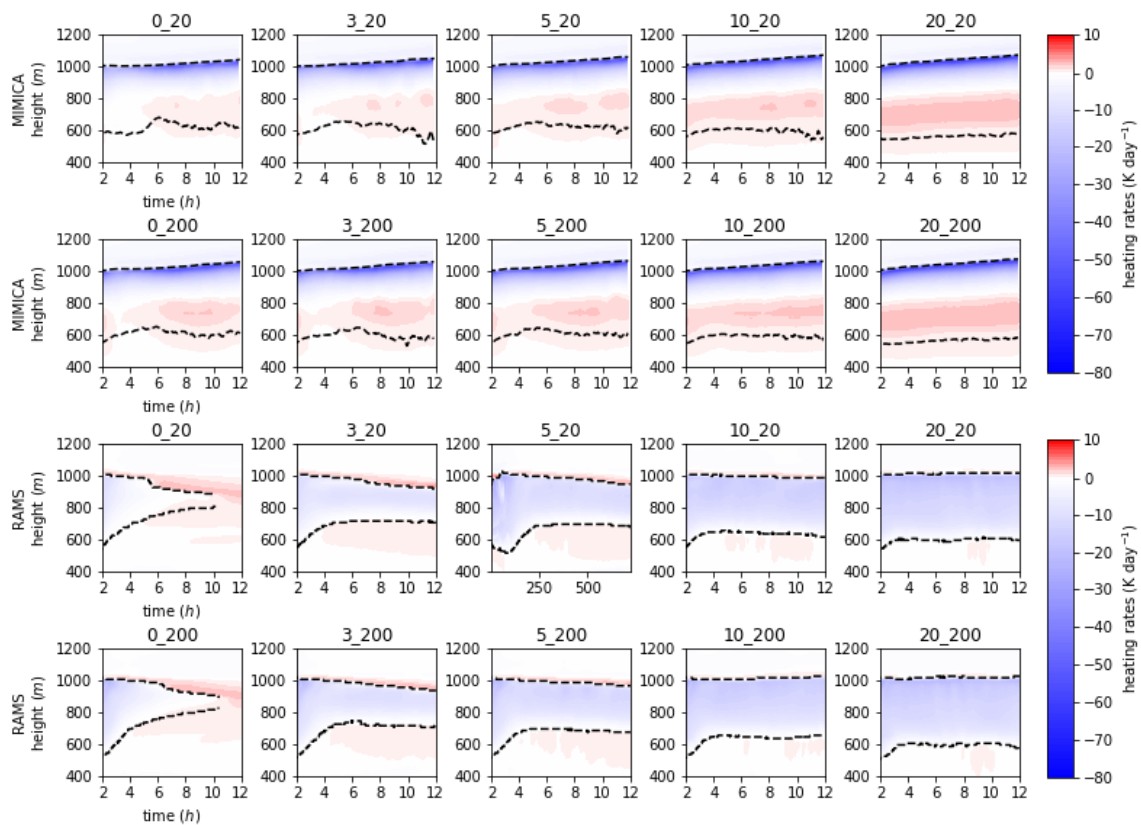

**Figure A2: Radiative heating rates for the MIMICA and RAMS simulation sets. The first 2 h of simulations, considered as spin-up, are excluded. For figure clarity, the plot titles have been abbreviated; the first number refers to the accumulation mode and the second to the Aitken mode concentration in cm$^{-3}$, i.e. "0_20" refers to "AC0_AK20". Black dashed lines represent the cloud top and cloud base heights.**

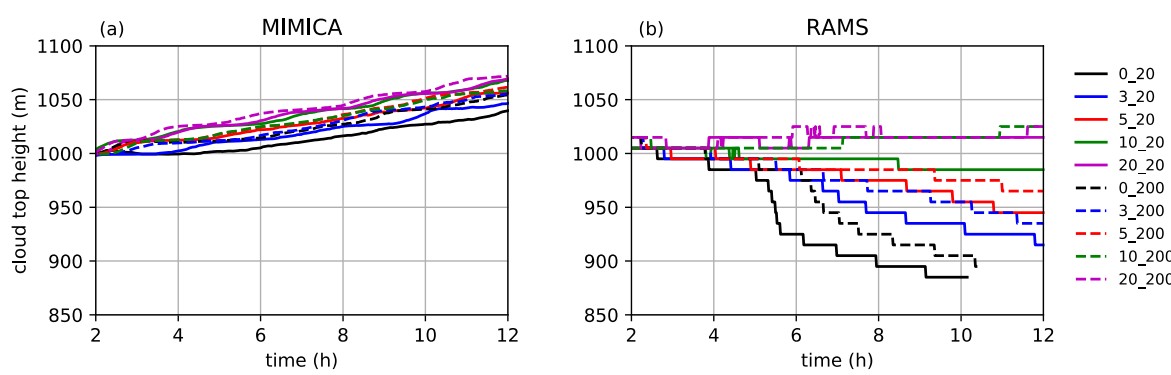

**Figure A3: Cloud top heights in (a) MIMICA and (b) RAMS. The first 2 h of simulations, considered as spin-up, are excluded.**

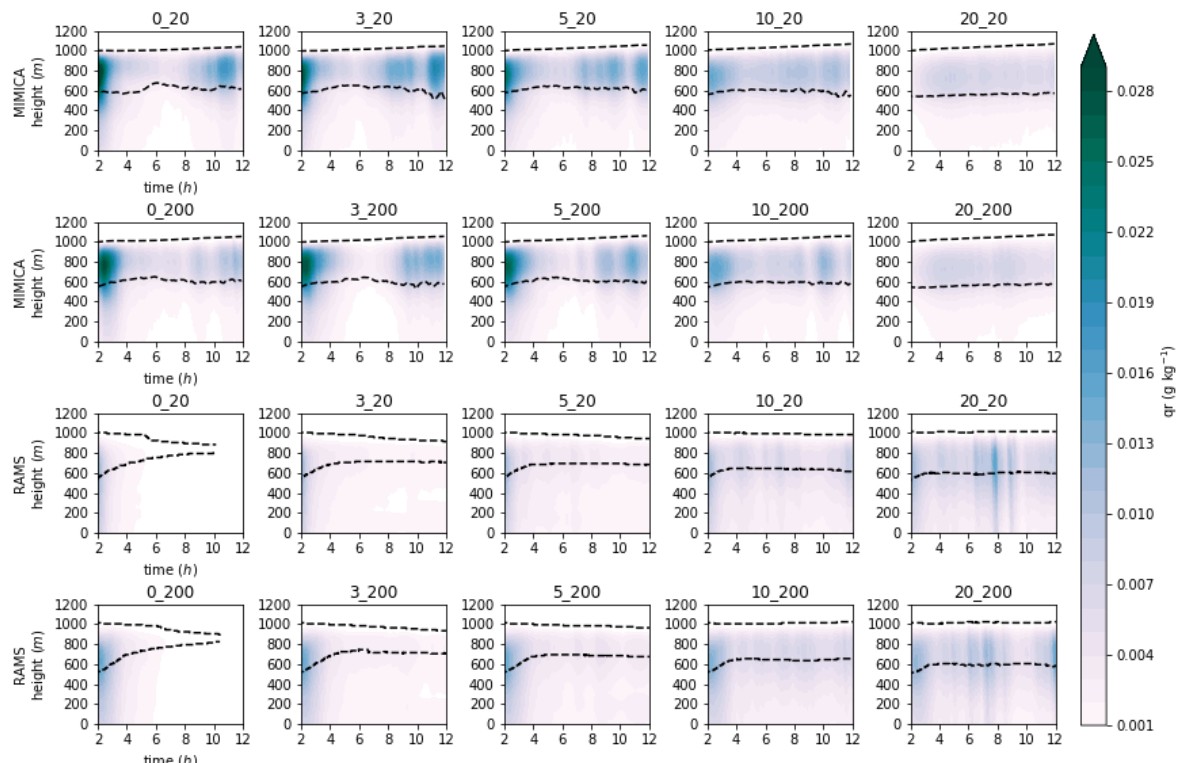

Figure A4: Rain mixing ratio (qr) for the MIMICA and RAMS simulation sets. The first 2 h of simulations, considered as spin-up, are excluded. For figure clarity, the plot titles have been abbreviated; the first number refers to the accumulation mode and the second to the Aitken mode concentration in cm$^{-3}$, i.e. "0_20" refers to "AC0_AK20". Black dashed lines represent the cloud top and cloud base heights.

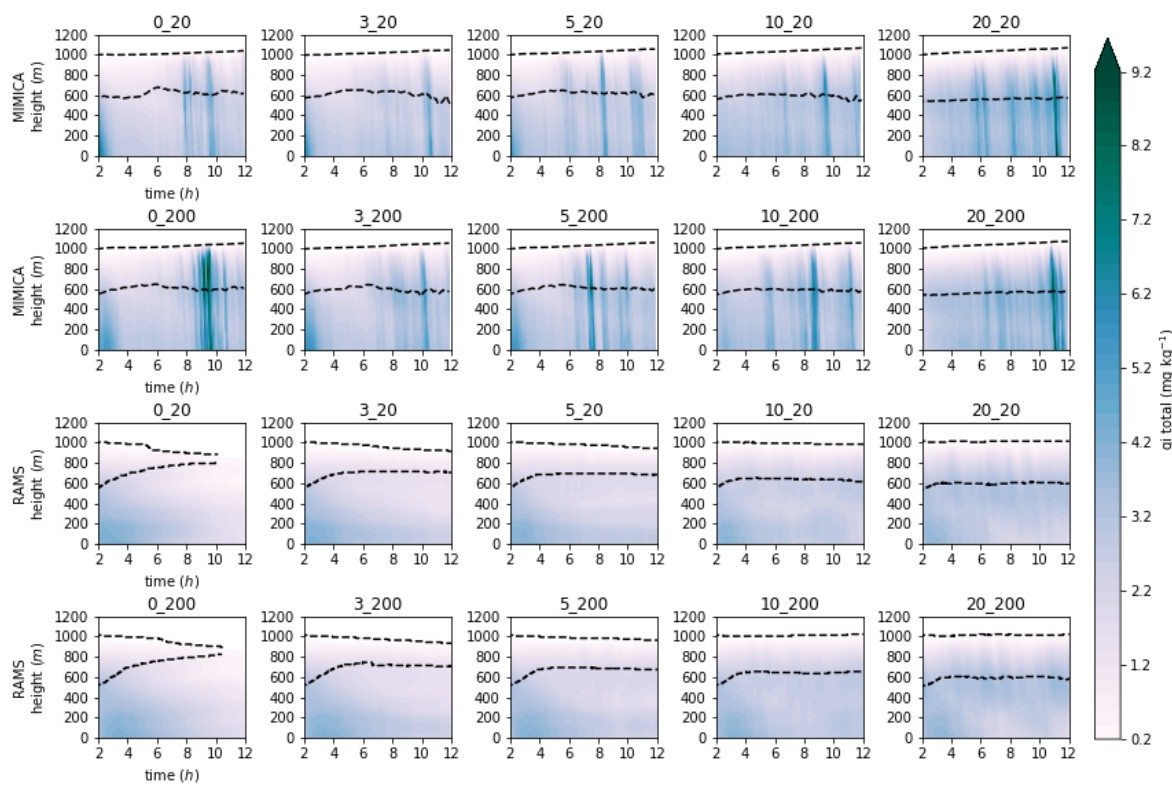

**Figure A5: Total ice mixing ratio (qi total) for the MIMICA and RAMS simulation sets. The first 2 h of simulations, considered as spin-up, are excluded. For figure clarity, the plot titles have been abbreviated; the first number refers to the accumulation mode and the second to the Aitken mode concentration in cm⁻³, i.e. "0_20" refers to "AC0_AK20". Black dashed lines represent the cloud top and cloud base heights.**

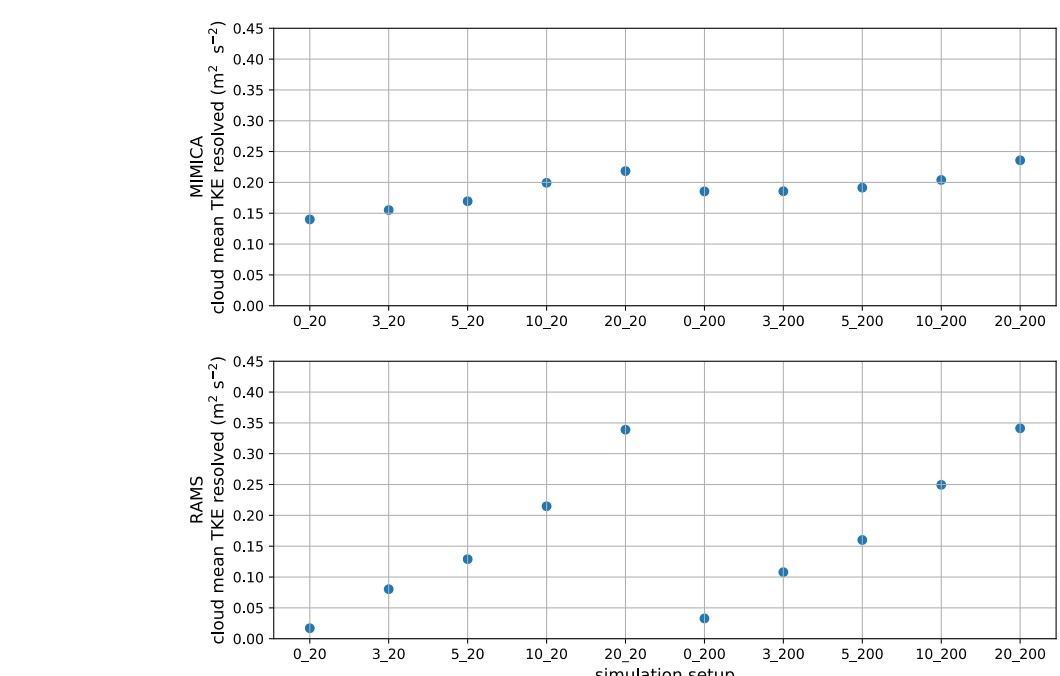

**Figure A6: Time-mean resolved turbulent kinetic energy (TKE) averaged for the cloud layer, simulated by MIMICA and RAMS. For figure clarity, the plot titles have been abbreviated; the first number refers to the accumulation mode and the second to the Aitken mode concentration in cm⁻³, i.e. "0_20" refers to "AC0_AK20".**






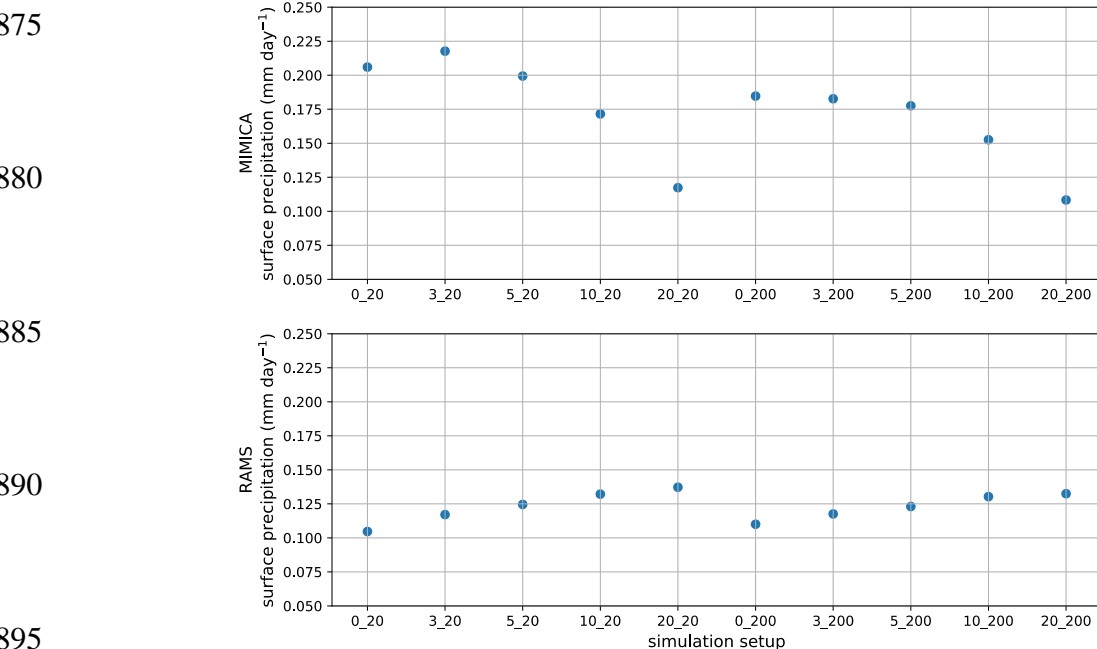

**Figure A7: Time-mean surface precipitation simulated by MIMICA and RAMS. For figure clarity, the plot titles have been abbreviated; the first number refers to the accumulation mode and the second to the Aitken mode concentration in cm⁻³, i.e. "0_20" refers to "AC0_AK20".**


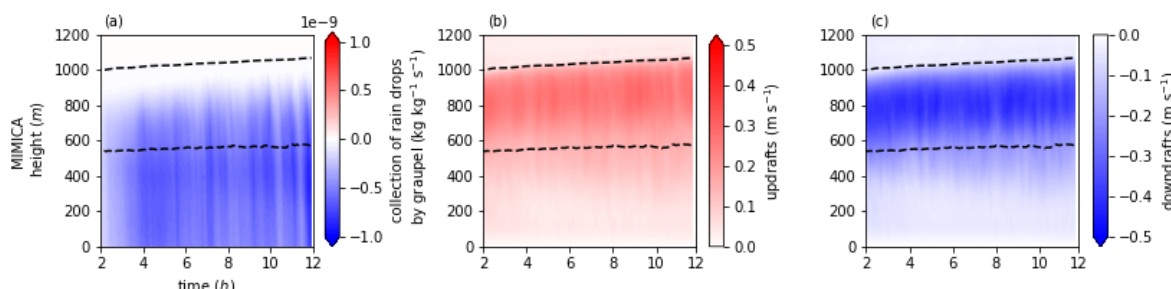


**Figure A8: (a) Collection of rain drops by ice (b) updrafts (c) downdrafts with time, simulated by MIMICA. The first 2 h of simulations, considered as spin-up, are excluded. For figure clarity, the plot titles have been abbreviated; the first number refers to the accumulation mode and the second to the Aitken mode concentration in cm⁻³, i.e. "3_20" refers to "AC3_AK20. Black dashed lines represent the cloud top and cloud base heights.**



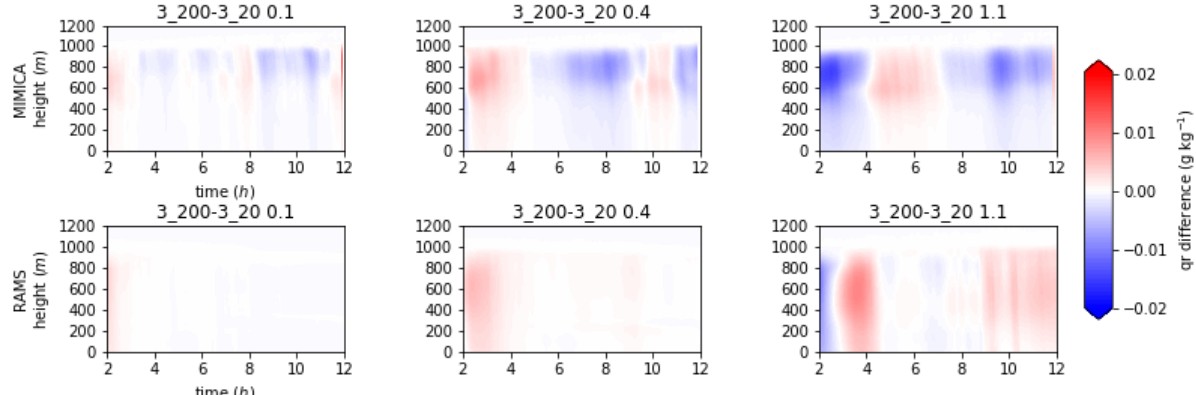

**Figure A9: Differences in rain mixing ratio (qr) for simulation pairs with the same accumulation mode concentration and the same kappa value of the Aitken mode particles: =0.1 (the leftmost column); =0.4 (the middle column); =1.1 (the rightmost column) shown for MIMICA and RAMS. The first 2 h of simulations, considered as spin-up, are excluded. A student's t-test with a 95% confidence level shows that the (time mean) differences are statistically significantly different for each pair of simulations. For figure clarity, the plot titles have been abbreviated; the first number refers to the**
**accumulation mode and the second to the Aitken mode concentration in cm⁻³, i.e. "3_20" refers to "AC3_AK20".**

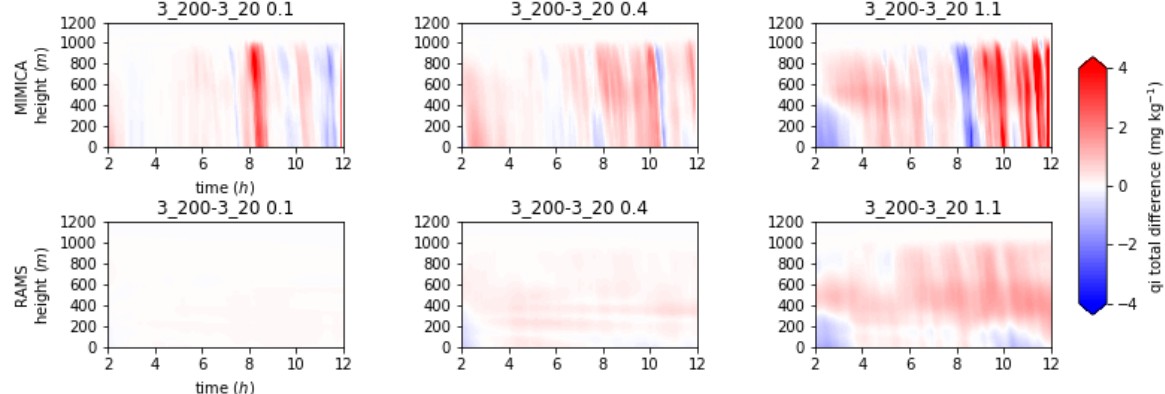


**Figure A10: Differences in total ice mixing ratio (qi total) for simulation pairs with the same accumulation mode concentration and the same kappa value of the Aitken mode particles: =0.1 (the leftmost column); =0.4 (the middle column); =1.1 (the rightmost column) shown for MIMICA and RAMS. The first 2 h of simulations, considered as spin-**
**up, are excluded. A student's t-test with a 95% confidence level shows that the (time mean) differences are statistically significantly different for each pair of simulations. For figure clarity, the plot titles have been abbreviated; the first number refers to the accumulation mode and the second to the Aitken mode concentration in cm⁻³, i.e. "3_20" refers to "AC3_AK20".**

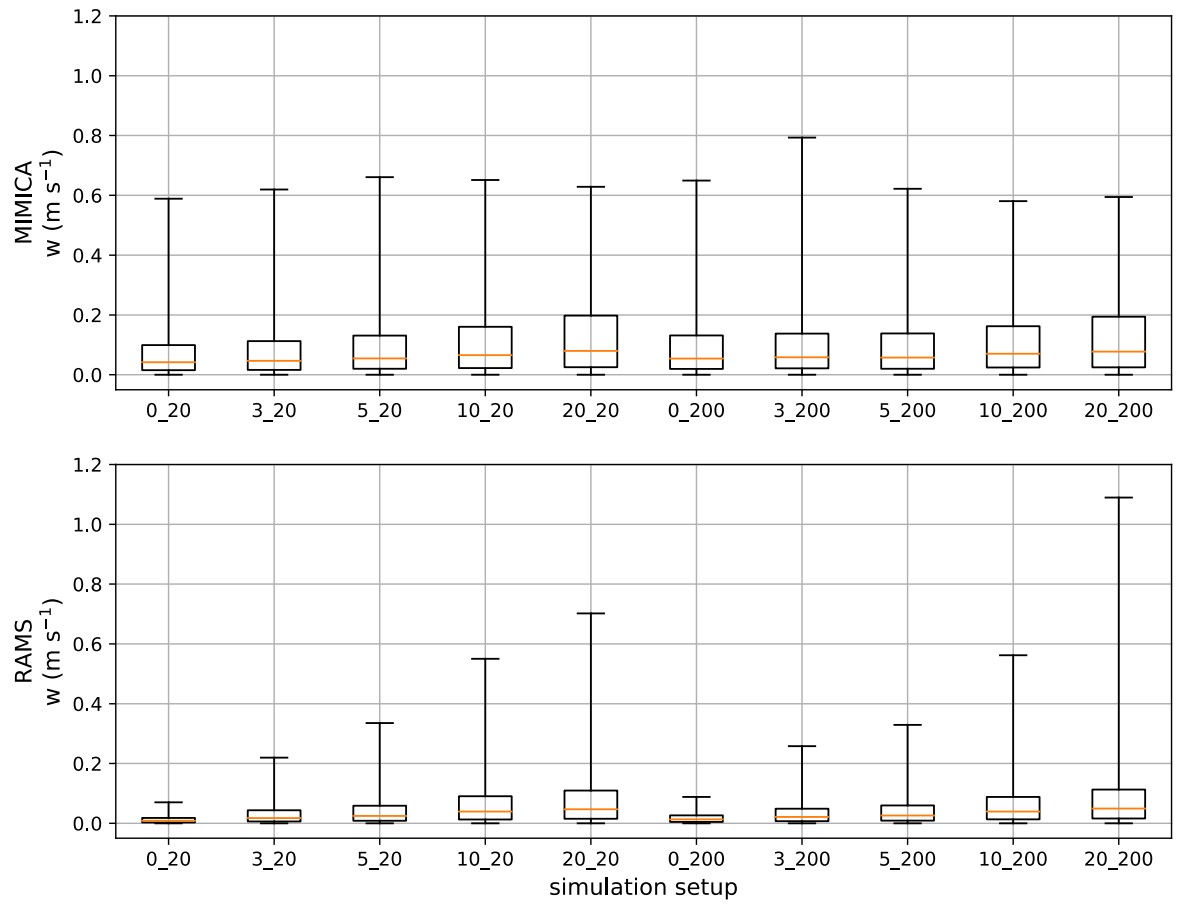


**Figure A11: Updraft (w) statistics simulated by MIMICA and RAMS. Lower and upper whiskers correspond to 1st and 99th percentiles, respectively. For figure clarity, the plot titles have been abbreviated; the first number refers to the accumulation mode and the second to the Aitken mode concentration in cm-3, i.e. "0_20" refers to "AC0_AK20".**


Figure A12 shows the relationship between critical supersaturation and dry diameters calculated for a range of kappa values, i.e. $\kappa$=[0.1; 0.2; 0.3; 0.4; 0.5; 0.6; 0.7; 0.8; 0.9; 1.0; 1.1]. The computation is done for the temperature T=298.15 K and the surface tension $\sigma s/a$=0.072 Jm$^{-2}$. More details on the calculations can be found in Petters and Kreidenweis (2007).


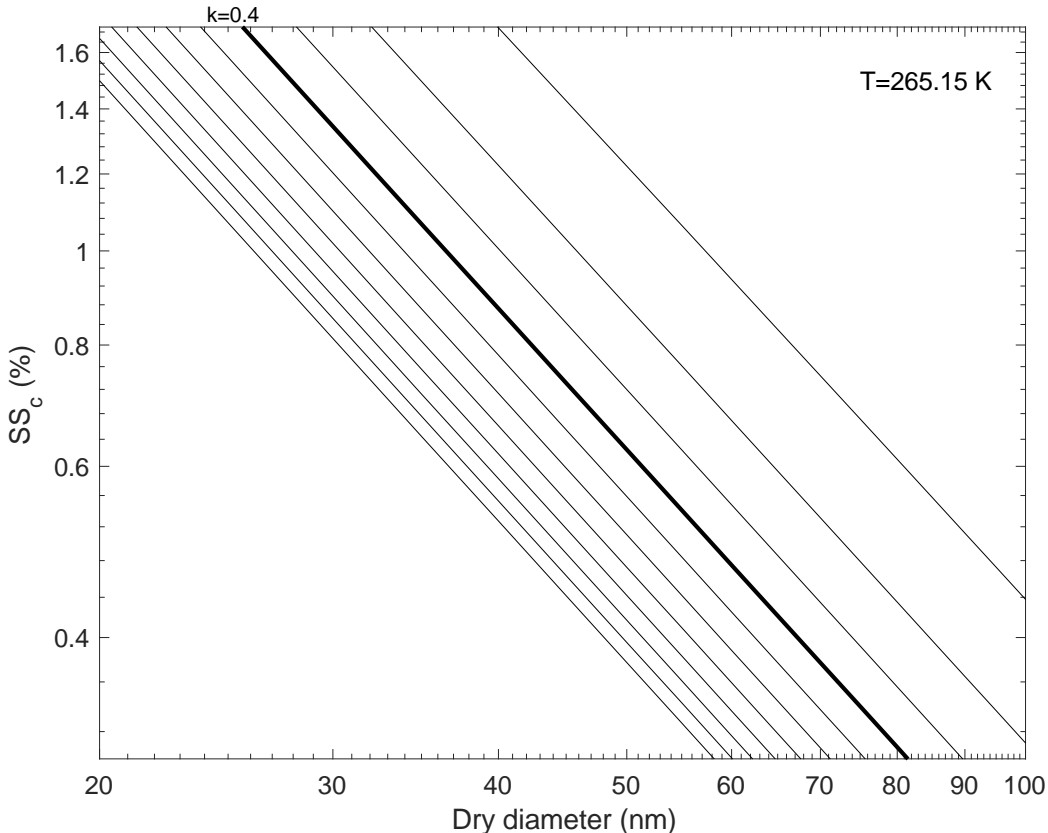

**Figure A12: Calculated critical supersaturations SSc (%) as a function of dry diameter, computed for σs/a=0.072 J m⁻² and T=298.15 K. κ-lines are shown for a range 0.1 ≤ κ ≤ 1.1. Bold line corresponds to kappa=0.4.**

*Data availability.* Modelling datasets used in this study are available at https://bolin.su.se/data/bulatovic-2020 with assigned doi (https://doi.org/10.17043/bulatovic-2020). The observational datasets are available from corresponding authors upon request.

*Author contribution.* IB, ALI and AMLE designed the experiments. IB and ALI performed the model simulations. IB analyzed the datasets. CL and JH provided the figures in the section "Qualitative comparison of model results with observational data for the High Arctic". IB prepared the manuscript with contributions from all co-authors.

*Competing interests.* The authors declare that they have no conflict of interest.

*Acknowledgements.* AMLE acknowledges the Swedish Science Foundation (Vetenskapsrådet), grant 2015-05318. ALI acknowledges the U.S. Department of Energy's Atmospheric System Research Grant DE-SC0019073. CL acknowledges Swedish Research Council (project no. 2016-03518, Leck). IR acknowledges Knut and Alice Wallenberg Foundation (Wallenberg Academy Fellowship project AtmoRemove 2015.0162) and European Commission (H2020 project FORCeS, grant agreement No 821205 and ERC-CoG project INTEGRATE, grant agreement No 865799). IB gratefully acknowledges Knut and Alice Wallenberg foundation for a scholarship from the Anniversary Travel Grants and the Bolin Centre Research group 2 for travel funds. The study was financially supported by the European Union's Horizon 2020 Research and Innovation Programme under grant agreement no. 641727 (PRIMAVERA).
The computations performed using MIMICA and data handling were enabled by resources provided by the Swedish National Infrastructure for Computing (SNIC) at the National Supercomputer Centre (NSC) partially funded by the Swedish Research Council through grant agreement no. 2016-07213.

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
