# Peer review of "The importance of Aitken mode aerosol particles for cloud sustenance in the summertime high Arctic: A simulation study supported by observational data"

_Atmospheric Chemistry and Physics, 2020_

## Referee Comment (RC1) · Anonymous Referee #1 · 21 Aug 2020

The study titled "**The importance of Aitken mode aerosol particles for cloud sustenance in the summertime high Arctic: A simulation study supported by observational data**" by Bulatovic et al. illustrates the impact of Aitken mode particles on summertime Arctic mixed-phase stratocumulus using a series of simulations by two different LES (RAMS and MIMICA). The authors show that Aitken mode particles significantly impact cloud microphysical particles and can contribute to cloud maintenance when accumulation mode particle concentrations are low. The reported results agree with observations from previous summertime campaigns in the Arctic and thus represent a realistic scenario for the high Arctic environment.

The manuscript is generally well written and contains an interesting combination of modeling and observational data. The study adds to our current understanding of aerosol-cloud interactions in Arctic mixed-phase clouds and highlights the importance of small-scale particles for mixed-phase cloud maintenance in the Arctic, which is relatively novel in this regard. Thus, the study has some implications regarding future model studies addressing the cloud response to aerosols in the summertime high Arctic. However, I have a few points that should be addressed before the manuscript is accepted for publication in ACP.

**General comments**

1. I am missing a section putting the findings of the study into perspective regarding previous work. A lot of studies have been published on aerosol-cloud interactions in the Arctic and the importance of CCN on cloud maintenance has been pointed out previously, especially for the ASCOS campaign (e.g., Loewe et al., 2017, Stevens et al., 2018), which is also simulated here. However, in previous work it was not distinguished between accumulation and Aitken mode aerosols, which is novel to the study presented here and should be pointed out more clearly.
Thus, it should also be emphasized more, when and where Aitken mode particles matter – here, a summertime mixed-phase cloud over pack ice is considered. Is the inclusion of Aitken mode particles only important in those clouds or also for clouds over the open ocean? What about other seasons? From the results presented here it seems like the results are exclusive to summer and pack ice, as in other seasons either accumulation mode aerosols are too numerous (e.g. spring), cloud ice is too high (e.g. winter), and over the ocean Aitken mode aerosols are less abundant; but if this is indeed the case it should be clearly highlighted and discussed.

2. Related to point 1: I would like to see the implications of including Aitken mode aerosol-cloud interactions on the local Arctic environment. As in summer, low-level clouds have an overall cooling effect on the surface which is important in terms of the ongoing Arctic warming, it would be interesting to see the implications on the energy balance at the surface, especially since Arctic Amplification and cloud-radiation interactions seem to be a motivation of this study as mentioned in the Introduction. Also, it would be interesting to see how surface precipitation changes, which could provide information on cloud maintenance beyond the simulated 12 h.

3. To me it is not clear why the model setup differs between the two models (which is especially important as there are quite some differences in the simulated cloud properties between the models). I assume there are reasons, but why is the vertical resolution different between both models? Throughout the manuscript, the authors point out the importance of entrainment and cloud top cooling for the cloud evolution in both models, however, the vertical resolution is essential in simulating these smaller-scale cloud top processes. Also, the inclusion of hail in RAMS can alter microphysical rates and cloud liquid and ice content, which is not discussed appropriately. Did the authors also perform simulations without including the hail category in RAMS?

4. The authors highlight the importance of Aitken mode particles for cloud sustenance in the title of this work and in the conclusions (line 576), however, in section 3.3.1 it is merely mentioned that Aitken mode particles have a significant impact on cloud droplet mixing ratio for up to 20 cm-3 of accumulation mode particles (RAMS) and 10 (MIMICA), but no statements about cloud sustenance are made. As this is an essential part of the paper, it should be pointed out in the results and be discussed more thoroughly (as mentioned above, also in terms of radiative impact, future implications, seasonal importance etc.).

5. The study has some caveats, which are not addressed at all but would be worth mentioning in a potential discussion section (maybe expand section 5 to a general discussion). Apart from some differing model settings as mentioned in point 3, the CCN are not prognostic, which certainly affects the cloud response to CCN. Similarly, the ice crystal number concentration is set constant, which also has implications for cloud properties in contrast to prognostic INPs such as used for example in Possner et al. (2017), Eirund et al. (2019) and Solomon et al. (2015, 2018). Lastly, secondary ice processes are omitted, however, I would imagine that they could play a role in summertime Arctic clouds as recently shown by Sotiropoulou et al. (2020).

**Specific comments**

**Abstract:**

**Line 18:** I find the expression "large-eddy simulation model" confusing as "simulation" already implies the term model. Maybe consider changing LES model (throughout the manuscript) with simply "LES", "models in LES mode" or similar.

**Line 27:** Related to my comment 1**,** it would be good to be more specific here in terms of when and where Aitken mode aerosols matter. You could add something like "for summertime MPCs over pack ice" (implying that this is when Aitken mode aerosols matter the most). If you additionally investigate the radiative response, it would be interesting to mention this here as well.

**Introduction:**

**Line 35**: The local lapse rate feedback has also been found to be important for Arctic Amplification, potentially even to be most important (e.g. Stuecker et al., 2018). Please add this here.

**Line 40:** The Arctic can also be in a persistent cloud-free state, thus consider changing "permanent" with "long-lived".

**Line 43:** "A layer of liquid is typically present at the top of SMP clouds" Please add a reference.

**Line 53:** The importance of free tropospheric humidity has also been shown by Solomon et al. (2011,2014) and Loewe et al. (2017).

**Line 59:** Consider linking these two paragraphs to point out that cloud microphysical properties and thus their radiative effect is for example impacted by aerosols…. (then go into introducing aerosols)

**Line 62:** There are actually a number of studies that have shown sensitivity of Arctic sea ice clouds to aerosols (e.g., Solomon et al., 2018, Stevens et al., 2018, Eirund et al., 2019).

**Line 65:** Also here there are more studies (e.g. the studies mentioned above) that have shown a strong impact of CCN changes to the radiative balance at the surface.

**Line 88:** "indirectly inferred" - this is very vague, what exactly did they show?

**Line 89**: From this section it is not clear to me what is different south of the ice edge as compared to over pack ice and between the Arctic and the high Arctic. Please be more specific in what previous studies have shown for which conditions and why more research is necessary.

**Line 102:** This is confusing to me. According to Koehler theory, large particles should always dominate the CCN availability as they activate more easily. Maybe change "even at low total aerosol concentrations" to "low accumulation mode aerosol concentrations"?

**Methods:**

**Line 121**: Are both of these models commonly used for simulating Arctic mixed-phase clouds? I have read several studies including MIMICA, but to me the RAMS model is rather uncommon for simulating Arctic clouds, so it would be good to add some references here that have used these models previously for similar studies.

**Line 168**: Here is one example (in addition to the introduction), where you point out the importance aerosol-cloud interactions for the surface energy budget, which is however never addressed. If you mention it as it is done here, it needs to be analyzed, otherwise please remove.

**Line 170**: According to the beginning of this section I assume the radiosonde was launched over sea ice, such that the surface conditions in the model are set to sea ice? If this is the case, please explicitly mention.

**Line 173**: "The cloud base and cloud top were nearly constant during the cloud lifetime (500 and 1000 m, respectively)." This implies to me that the cloud has been observed over a longer time period, however, in Figure 1 it looks like the observations show only one point in time. Please clarify and/or change the layout of Figure 1 (see my comment regarding Figure 1).

**Line 196**: "The concentrations are chosen to cover typical aerosol size distributions often encountered in the summertime high Arctic (Heintzenberg and Leck, 2012; Leck and Svensson, 2015)." – for both, accumulation and Aitken mode aerosols?

**Line 197**: Better use "assumed" rather than "considered"?

**Line 200**: As you performed quite a large set of simulations, if would be helpful for the reader to have an additional table including all simulations and the varying type/number of aerosol and model that could be referred to here. Also simulation AC20_AK20 could be clearly marked as baseline or control simulation.

**Line 203**: "fluxes were small" – how small (please be specific)? Is a prescribed flux of zero justified?

**Same line**: Is the choice of 0.844 for surface albedo arbitrary or is there a reference?

**Same paragraph**: Was there large-scale advection? And how was the roughness length defined? Also I assume the surface condition was set to sea ice? Please add these information to the simulation setup.

**Line 210**: Based on what conditions was a spin-up of 2 h chosen?

**Line 229**: Why were these simulations only performed with MIMICA?

**Line 232**: "relatively ice-free" An ice crystal number concentration of 0 L-1 is completely ice free, not only relatively ice-free. I would remove "relatively".

**Results:**

**Figure 1**: As mentioned above, it looks like you are comparing the temporal evolution of the modeled clouds with an observed temporal snapshot (if this is the case). It would be helpful to have more information about the observations (also how are the percentiles derived, are the observations constant in time?) in the text and the Figure caption. Also it would be interesting to know, if the models and the observations cover the same vertical range of the cloud (if these information are available from the observations). You could consider changing the layout of this figure to height on the y-axis and cloud liquid water and ice content over height on the x-axis.

**Line 255**: It looks to me that RAMS has a higher autoconversion, as also qr is higher (as shown in Figure 2). If this is indeed the case, maybe mention autoconversion as an additional point.

**Line 259**: Could another reason be the inclusion of hail in RAMS? Does hail maybe increase surface precipitation, which then reduces the overall LWP and IWP in RAMS? Have you tried to switch hail off?

**Line 265**: Is it possible that the additional rain formation in RAMS stabilizes the cloud and prevents a continuous cloud top rise as seen in MIMICA?

**Line 266**: Where do these peaks in qi come from? I would expect to see corresponding peaks for example in the radiative cooling rate, which would hint towards enhanced growth by deposition at certain times, but I cannot find any evidence there.

**Line 270**: Entrainment can also lead to drying (Ackermann et al., 2004) - was there a moist layer present in the observations? Maybe it would be good to show the initial profiles as measured by the radiosonde to see the temperature and specific humidity (or total moisture) vertical distributions? Also is looks like there is enhanced evaporation of cloud droplets just above cloud top (Figure 3a,b), which would hint towards a drier layer overlying the cloud? In this case entrainment would rather dry than moisten the cloud.

**Line 278**: Is the correlation of condensation with updrafts shown anywhere? Or is this a general statement? In the latter case please add a reference.

**Figure 3**: Please consider changing the colorscale. I understand that smaller values in MIMICA have to be represented, but for example in Figure 3d or f I cannot see the total magnitude at all.

**Line 329**: "thins" rather than "shrinks"

**Line 334 and Figures 5 and A2**: Is this the cooling or the heating rate (such that neg. values indicate a cooling as in Brooks et al. (2017) which is what I assume following your arguments)? Also the differing colorscales are very confusing, especially since the numbers are very small and an additional zero can easily been overseen. Please consider changing the units to K/h or K/d and try to keep the colorscales the same. As the cloud top is essential for the analysis, it might be worth zooming into the cloud top to better resolve the magnitude of the cooling there. You could also add cloud top such as shown in Figure A3 within Figure 5, so the reader can easily identify cooling at the cloud top.

**Line 337**: In line 253 you say that stronger radiative cooling rates in MIMICA produce a higher LWP, while here you state that the higher qc leads to stronger cooling in MIMICA. Of course it's a feedback where high LWP changes the cloud emissivity, which in turn increases longwave cooling which then

again favors enhanced turbulence and condensation. However, above a certain threshold of approximately 40 g m-2 the sensitivity of longwave cooling to LWP becomes small (e.g. Garret and Zhao, 2006) which is why I assume you point out the difference in cloud top liquid water here. Please be more specific in your line of argumentation of the qc/cloud top cooling feedback and also refer to other studies of Arctic mixed-phase clouds which have investigated this correlation as well.

**Line 343**: Not only the entrainment of moist air, but also the increase in vertical motions as a result of more turbulence can favor condensation and maintain cloud liquid (Shupe et al., 2008).

**Line 347**: As mentioned in my point 3, does maybe the different vertical resolution also play a role here as well?

**Figure 4**: It looks like the high end of the colorscale is never reached, thus consider adjusting it.

**Line 380**: I see a very strong signal of Aitken mode particles for low number concentrations of accumulation mode particles, which underlines the main message of the manuscript (cloud maintenance for 6 h). Why do you say there no clear trend?

**Line 387**: Here you say there is a significant impact, while above you write there is no clear trend (see my previous comment). I would agree with this statement and emphasize it more, as it is one of the main messages of the manuscript.

**Line 410**: why do more Aitken mode particles lead to more turbulence? Because of more latent heat release through condensation? If this is the case, please clearly state it.  However, the updrafts for low accumulation mode and different Aitken mode particles look very similar, but this could be due to temporal averaging I assume.

**Line 415/Figure A9**: I assume Figure A9 shows spatial and temporal statistics? In this case I cannot see a temporal evolution in the updrafts that would determine the temporal evolution of rain. Please make this argument clearer.

**Line 418**: Previous studies have also shown an impact of CCN on cloud ice and increased LW cooling of CCN-perturbed clouds has been identified as driving force for increased immersion freezing and growth by deposition (Possner et al., 2017, Solomon et al., 2018, Eirund et al., 2019). As the ice crystal number concentrations is fixed, immersion freezing does not play a role here, but I would suspect that LW cooling does also impact growth by deposition and thus qi.  This might be worth exploring/mentioning.

**Line 426**: This is an interesting finding. The effect of CCN changes for different background ice crystal number concentrations/INP concentrations has been studied previously (Stevens et al., 2018, Possner et al., 2017). There, also a smaller CCN impact was found for higher ice crystal number concentrations /INPs, which would agree with your findings and might be worth mentioning.

**Supersaturation statistics:**

**Line 487:** Why did you choose a 20 min interval around 6 h of simulation time? From Figure 6 it looks like strongest signals in qc are either in the beginning or towards the end of the simulations (in RAMS); is your choice of time period linked to any criteria?

**Line 515**: Again, previous studies have also shown an increase in cloud-top radiative cooling in seeded clouds (see my previous comment). Please cite previous studies accordingly.

**Qualitative comparison of model results with observational data for the High Arctic:**

**Line 338/Section 5**: This section is very interesting and gives the authors the opportunity to make statements about the relevance of their work in a broader scope. However, this section is relatively short in my opinion and could be expanded. Especially, it would be interesting to know when/where is including Aitken mode particles important and should be considered in modeling studies? Also the authors could expand section 5 to a general discussion of the results and could compare their findings to previous work which investigated the response of Arctic cloud to varying CCN concentrations.

**Summary and conclusions:**

**Line 580:** "Aitken mode aerosols have a significant impact on the cloud droplet amount" **-** All Figures show differences in mixing rations, not number concentrations (hence cloud droplets could also be larger, not necessarily more numerous). Did I miss something or should this be "cloud liquid water"? I agree that the increase in qc most likely results from an increase in Ndrop, but this has not been shown, as far as I can see.

**Line 582**: Again "higher number of cloud droplets", same comment as above.

**References**

Ackermann et al., 2004: The impact of humudity above stratiform clouds on indirect aerosol climate forcing, Nature, 432, 1014-1017, doi: 10.1038/nature03137.1

Eirund et al., 2019: Response of Arctic mixed-phase clouds to aerosol perturbations under different surface forcings, Atmos. Chem. Phys., 19, 9847–9864, doi: 10.5194/acp-19-9847-2019

Garret and Zhao, 2006: Increased Arctic cloud longwave emissivity associated with pollution from mid-latitudes, Nature, 440, 787-789, doi: 10.1038/nature04636

Loewe et al., 2017: Modelling micro- and macrophysical contributors to the dissipation of an Arctic mixed-phase cloud during the Arctic Summer Cloud Ocean Study (ASCOS), Atmos. Chem. Phys., 17, 6693-6704, doi: 10.5194/acp-2016-917

Possner et al., 2017: Cloud response and feedback processes in stratiform mixed-phase clouds perturbed by ship exhaust, Geophys. Res. Lett., 44, 1964-1972, doi: 10.1002/2016GL071358

Shupe et al., 2008: Vertical Motions in Arctic Mixed-Phase Stratiform Clouds, J. Atmos. Sci., 65, 1304-1322, doi: 10.1175/2007JAS2479.1

Solomon et al., 2011: Moisture and dynamical interactions maintaining decoupled Arctic mixed-phase stratocumulus in the presence of a humidity inversion, Atmos. Chem. Phys., 11, 10127-10148, doi: 10.5194/acp-11-10127-2011

Solomon et al., 2014: The Sensitivity of Springtime Arctic Mixed-Phase Stratocumulus Clouds to Surface-Layer and Cloud-Top Inversion-Layer Moisture Sources, J. Atmos. Sci., 71, 574-595, doi: 10.1175/JAS-D-13-0179.1

Solomon et al., 2015: The role of ice nuclei recycling in the maintenance of cloud ice in Arctic mixed-phase stratocumulus, Atmos. Chem. Phys., 15, 10631-10643, doi: 10.5194/acp-15-10631-2015

Solomon et al., 2018: The relative impact of cloud condensation nuclei and ice nucleating particle concentrations on phase-partitioning in Arctic Mixed-Phase Stratocumulus Clouds, Atmos. Chem. Phys., 18, 17047-17059, doi: 10.5194/acp-2018-714

Sotiropoulou et al., 2020: The impact of secondary ice production on Arctic stratocumulus, Atmos. Chem. Phys., 20, 1301-1316, doi: 10.5194/acp-20-1301-2020

Stevens et al., 2018: A model intercomparison of CCN-limited tenuous clouds in the high Arctic, Atmos. Chem. Phys., 15, 11041-11071, doi: 10.5194/acp-18-11041-2018

Stuecker et al., 2018: Polar amplification dominated by local forcing and feedbacks, Nature Climate Change, 8, 1076-1081, doi: 10.1038/s41558-018-0339-y

---

## Referee Comment (RC2) · Anonymous Referee #2 · 26 Aug 2020

General comments:

The authors employed two numerical models to perform a series of simulations to study the importance of Aitken mode aerosol particles for cloud sustenance in the summertime high Arctic. The messages in the abstract seem to be clear. After reading through the main text of the paper, I find some interesting results. But I am also confused by arbitrary model and simulation configurations and overwhelmed by poorly interpreted, disorganized, and probably unnecessary results. Throughout the manuscript, the authors used a lot of speculations in their reasoning where solid evidences are expected.

[Figure]

The writing also has huge room for improvement. I listed my major and minor concerns as well as suggestions regarding the technical aspects of the manuscript below.

Specific comments (major):

If I understand correctly, the main idea of this paper is that, for some combinations of model configurations (i.e., Aitken mode and accumulation mode aerosol number concentration, aerosol kappa value, and ice number concentration), the modeled cloud can survive through 12 h, meaning that during this period, the clouds can maintain sufficient liquid water against processes that depletes liquid water (like subsidence, entrainment of warm air, losing moisture due to glaciation and precipitation, so on) and generate enough supersaturation to activate the prescribed Aitken mode aerosol. Was the specified aerosol size distribution for each simulation used from the beginning of the simulation? How important are aerosols during the spin-up? In other words, are the differences among simulations using the same model and between MIMICA and RAMS due to the activation behavior when the turbulent motion is very weak? What if all simulations (in each model) begin with a robust cloud and fully-developed the turbulence, spun-up using same configurations, and then switch to different aerosol number concentrations? I don't think a juicy but non-turbulent cloud is a realistic starting point to test aerosols' impacts on sustaining clouds or produce results that are relevant to the real world. Please justify this choice.

A few other questions related to initialization and spin-up: Is the initial cloud size distribution related to the Aitken mode and accumulation mode aerosol? For each model, are the liquid water content profiles in all simulations identical at the beginning? Are all microphysical processes (e.g., all processes related to precipitations) turned on from the beginning?

It is worthwhile to be more specific about the activation of aerosol particles in MIMICA and RAMS as configured for this study. It seems that the activation scheme in MIMICA is identical to the one used to generate Fig. A10. What about in RAMS? Are the ss in

Fig. 10 and Fig. 11 same as those used in the activation scheme in the models?

It seems that the authors tried to use an observed sounding to set up the baseline simulations, and then perform sensitivity tests on top of that. However, the authors did not provide enough details (for example, dedicated figures) for readers to understand the case. Please consider showing some details. I found the ASCOS sounding available from https://bolin.su.se/data/ascos-radiosoundings. Did the authors use the original 0535 UTC 31 August 2008 sounding from this archive? Or an idealized version of it? The authors used the observed CCN to justify the use of AC20_AK20 as the baseline simulation. However, the cloud layer in the aforementioned sounding seems to be decoupled from the surface. If this is the case, does it still make sense to use surface measurement to determine the base case?

The authors mentioned a few times that Arctic stratocumulus may entrain moist air from above the cloud top, but provided no evidence. (Whether other studies showed entrainment of moisture from above the cloud could happen is irrelevant to this study.) Initial sounding (together with profiles from the middle of the simulations) can be used to show whether there is moist layer above the cloud top.

Choice of model configurations, shared simulation setup, and experiment design are perplexing and arbitrary. Why were the vertical resolutions different, especially as the authors suspected that the different vertical resolutions may be the source of some discrepancies between the simulation results from the two models (e.g., L254). Why were the microphysics in RAMS so much more complicated (even with hail turned on) than in MIMICA? Why was aggregation turned off and only for MIMICA? Why was MIMICA used for ice number concentration sensitivity test while certain ice-related budget terms are not available from it (L289)?

Much of the results regarding rain and ice are only superficially described and discussed, with no obvious connection to the main goal of the paper, and sometimes contain errors. A few examples are provided here.

L277: "The pockets of condensation and evaporation present in the main cloud layer are well-correlated with updrafts and downdrafts and they tend to cancel each other in the mean. This is why the average condensation rate in the main cloud is of the same order of magnitude as the one in the sub-cloud layer." Totally lost.

L415: "The presence of both positive and negative differences with time is a result of differences in cloud dynamics with different distributions of updrafts and downdrafts with time that govern the rain production in the cloud (cf. Fig. A9)." But there is nothing about "distributions with time" in Fig. A9.

L421: "change in the Aitken mode particle number concentration results in that maximum updrafts are reached at somewhat different times", what is the significance of this?

L422: "Differences are in general greater in MIMICA than in RAMS since there is a slightly more total ice in MIMICA". Does "since" mean "because" here? Does greater total ice water content (or path?) have to lead to greater differences?

Other than the "entrainment of moisture aloft", here are a few additional examples of unacceptable speculation.

L288-289, "Examining the total ice deposition/sublimation rates would most likely lead to similar rates between the two models". (BTW, "The ice crystal deposition and sublimation rates are higher in RAMS than in MIMICA since the two models partition the total ice deposition differently among ice hydrometeor categories" is also just speculation, isn't it?)

L411-412: "an increase in the Aitken mode particle concentration may lead to stronger turbulence and more cloud liquid water production". Turbulence intensity and cloud liquid water budget can be diagnosed. It does not make sense to speculate ("may lead to").

Minor comments:

Contents in Sections 2.1 and 2.3 are not clearly separately. For example, why are number of model grid points and resolution introduced in Section 2.1 but domain size in Section 2.3?

Please clarify a few model or simulation configurations issues: Is it correct that the time step is 2 s for MIMICA? What about RAMS? The terminal speed is only introduced for MIMICA, what about RAMS? What are the references for the terminal speed formulas used? Is the 0.2 L-1 ice number concentration based on ASCOS observations? Need reference. Why is divergence set to 1.5E-6 s-1? Is the same value used for all model layers?

L130: Is the wet deposition calculated for any tracers in either model?

L346: Since they are simplified LW cooling calculation that only depends on LWC, why not using same formula for both MIMICA and RAMS?

L349: "Stable cloud base", do you mean "steady"?

L383: "Fig. 5", should be Fig. 6?

Examples of poor writing:

L265: "total ice", do you mean "total ice mixing ratio"? There are a few "total ice" throughout the text.

L377: "available water vapor" what is this?

Suggestions on technical issues:

Please consider marking cloud top and base whenever relevant.

Colors are saturated in some figures, e.g., Fig. 3f. Please adjust.

L292: "from the different model descriptions", should be "configurations"?

---

## Author Comment (AC1) · 2 Nov 2020

The study titled "The importance of Aitken mode aerosol particles for cloud sustenance in the summertime high Arctic: A simulation study supported by observational data" by Bulatovic et al. illustrates the impact of Aitken mode particles on summertime Arctic mixed-phase stratocumulus using a series of simulations by two different LES (RAMS and MIMICA). The authors show that Aitken mode particles significantly impact cloud microphysical particles and can contribute to cloud maintenance when accumulation mode particle concentrations are low. The reported results agree with observations

from previous summertime campaigns in the Arctic and thus represent a realistic scenario for the high Arctic environment.Âă

The manuscript is generally well written and contains an interesting combination of modeling and observational data. The study adds to our current understanding of aerosol-cloud interactions in Arctic mixed-phase clouds and highlights the importance of small-scale particles for mixed-phase cloud maintenance in the Arctic, which is relatively novel in this regard. Thus, the study has some implications regarding future model studies addressing the cloud response to aerosols in the summertime high Arctic. However, I have a few points that should be addressed before the manuscript is accepted for publication in ACP.Âă

We thank the reviewer for his/her careful reading of the manuscript and many constructive comments.

General commentsÂă

Âă1. I am missing a section putting the findings of the study into perspective regarding previous work. A lot of studies have been published on aerosol-cloud interactions in the Arctic and the importance of CCN on cloud maintenance has been pointed out previously, especially for the ASCOS campaign (e.g., Loewe et al., 2017, Stevens et al., 2018), which is also simulated here. However, in previous work it was not distinguished between accumulation and Aitken mode aerosols, which is novel to the study presented here and should be pointed out more clearly.Âă Thus, it should also be emphasized more, when and where Aitken mode particles matter – here, a summertime mixed-phase cloud over pack ice is considered. Is the inclusion of Aitken mode particles only important in those clouds or also for clouds over the open ocean? What about other seasons? From the results presented here it seems like the results are exclusive to summer and pack ice, as in other seasons either accumulation mode aerosols are too numerous (e.g. spring), cloud ice is too high (e.g. winter), and over the ocean Aitken mode aerosols are less abundant; but if this is indeed the case it should be clearly

highlighted and discussed.Âă

We have now expanded the text on previous studies that have investigated the CCN influence on high Arctic clouds and the seasonal cycle of aerosol particles in this region in Section 1 and 5. We have also contrast areas over the pack ice and open ocean. In this way we pointed out why Aitken mode particles can be particularly important for the summertime low-level SMP clouds over the Arctic pack ice area. Section 5 includes new Subsection: 5.2. General importance of Aitken mode particles for low-level mixed-phase cloud properties.

2. Related to point 1: I would like to see the implications of including Aitken mode aerosol-cloud interactions on the local Arctic environment. As in summer, low-level clouds have an overall cooling effect on the surface which is important in terms of the ongoing Arctic warming, it would be interesting to see the implications on the energy balance at the surface, especially since Arctic Amplification and cloud-radiation interactions seem to be a motivation of this study as mentioned in the Introduction. Also, it would be interesting to see how surface precipitation changes, which could provide information on cloud maintenance beyond the simulated 12 h.Âă

We thank the reviewer for this important input and have now added a figure that shows the differences in downward LW radiation at the surface between the cases with the same number of accumulation mode particles but with different Aitken mode concentrations (Figure 10). Differences in SW radiation were found to be small and are only discussed in the text. We have also added the surface precipitation figure in the Appendix.

3. To me it is not clear why the model setup differs between the two models (which is especially important as there are quite some differences in the simulated cloud properties between the models). I assume there are reasons, but why is the vertical resolution different between both models? Throughout the manuscript, the authors point out the importance of entrainment and cloud top cooling for the cloud evolution in both models, however, the vertical resolution is essential in simulating these smaller-scale cloud top processes. Also, the inclusion of hail in RAMS can alter microphysical rates and cloud liquid and ice content, which is not discussed appropriately. Did the authors also perform simulations without including the hail category in RAMS?Âă

Our aim was to use the default setup for both models, as this is what typically would be used for an arbitrary study. For example, modelling studies that use only one model to compare with observations would most likely use the default model setup. The default setup in RAMS is a fixed vertical grid spacing while in MIMICA the default is a variable grid spacing. Nevertheless, we agree with the reviewer that the difference in vertical grid spacing could be an important reason for the simulated differences in cloud properties. Therefore, we ran an additional MIMICA simulation for the baseline case with a fixed vertical grid spacing of 10 m like in RAMS. The test did not show any significant difference in the simulated cloud microphysical properties compared to the MIMICA baseline case with a variable grid spacing. Thus, the difference in resolution at cloud top does not impact our conclusion. The inclusion of hail is also a standard setting in RAMS. We agree that hail generally can change microphysical rates, however, in all RAMS simulations 97.9-99.7% of the ice is present as ice crystals so the riming treatment plays a very small role in these simulations. We have also checked the hail contribution to the total surface precipitation rates and it is 2 order of magnitude less than the contribution from rain. The differences in model setup in terms of vertical resolution and hail are now discussed in the manuscript (Subsect. 2.3).

4. The authors highlight the importance of Aitken mode particles for cloud sustenance in the title of this work and in the conclusions (line 576), however, in section 3.3.1 it is merely mentioned that Aitken mode particles have a significant impact on cloud droplet mixing ratio for up to 20 cm-3 of accumulation mode particles (RAMS) and 10 (MIMICA), but no statements about cloud sustenance are made. As this is an essential part of the paper, it should be pointed out in the results and be discussed more thoroughly (as mentioned above, also in terms of radiative impact, future implications, seasonal

importance etc.).Âă

We agree with this point. We have now added a statement about the cloud sustenance in Sect. 3.3.1. In the same section, we have also added a figure that shows the influence of Aitken mode particles on the downward LW radiation at the surface (Fig. 10). The seasonal importance and future implications are discussed in the Sect. 5.2 and 6.

5. The study has some caveats, which are not addressed at all but would be worth mentioning in a potential discussion section (maybe expand section 5 to a general discussion). Apart from some differing model settings as mentioned in point 3, the CCN are not prognostic, which certainly affects the cloud response to CCN. Similarly, the ice crystal number concentration is set constant, which also has implications for cloud properties in contrast to prognostic INPs such as used for example in Possner et al. (2017), Eirund et al. (2019) and Solomon et al. (2015, 2018). Lastly, secondary ice processes are omitted, however, I would imagine that they could play a role in summertime Arctic clouds as recently shown by Sotiropoulou et al. (2020).Âă

Please see our reply on point 3 regarding the different model configurations. The number of CCN is actually prognostic in the study as it is calculated from the prescribed aerosol size distributions. However, there is no sink or source of aerosols. We agree with the reviewer that if there would be the sink, then the effect of Aitken mode particles would be most likely even larger since accumulation mode particles are larger in size and thus they would be removed more easily. This is now mentioned in the new version of the manuscript (Subsect. 5.2). We also agree with the reviewer that the prognostic INPs could give a higher sensitivity compared to prescribed ice crystal concentrations. This is now discussed in the Discussion subsection 5.2. The aggregation of ice crystals is actually included in both models. The previous statement that aggregation is omitted in MIMICA was a mistake and has been corrected.

Specific commentsÂă

Abstract:Âă Line 18: I find the expression "large-eddy simulation model" confusing as "simulation" already implies the term model. Maybe consider changing LES model (throughout the manuscript) with simply "LES", "models in LES mode" or similar.Âă

We respectfully disagree with the reviewer on this point and argue that "LES model" can be used. MIMICA and RAMS are models that utilize large-eddy simulation (as a technique). We have thus kept the term "LES model" in the manuscript.

Line 27: Related to my comment 1, it would be good to be more specific here in terms of when and where Aitken mode aerosols matter. You could add something like "for summertime MPCs over pack ice" (implying that this is when Aitken mode aerosols matter the most). If you additionally investigate the radiative response, it would be interesting to mention this here as well.Âă

The first sentence in the abstract was: "The potential importance of Aitken mode particles (diameters ∼25–80 nm) for stratiform mixed-phase clouds in the summertime high Arctic has been investigated using two large-eddy simulation models". We have now added: "...summertime high Arctic (> 80 N) ..." to be more specific what term "high" means. We have also added that the Aitken mode particles have a significant impact on radiative properties of the cloud (which is shown in the Results section).

Introduction:Âă Line 35: The local lapse rate feedback has also been found to be important for Arctic Amplification, potentially even to be most important (e.g. Stuecker et al., 2018). Please add this here.Âă

We agree with the reviewer. The local lapse-rate feedback has been added in parenthesis as one local feedback. The reference is also added.

Line 40: The Arctic can also be in a persistent cloud-free state, thus consider changing "permanent" with "long-lived".Âă

Indeed, this was a mistake. "permanent" has been changed with "persistent".

Line 43: "A layer of liquid is typically present at the top of SMP clouds" Please add a

reference.Âă

References have been added.

Line 53: The importance of free tropospheric humidity has also been shown by Solomon et al. (2011,2014) and Loewe et al. (2017).Âă

In the previous version of the manuscript, this paragraph was focused on low-level SMP clouds in the high Arctic (> 80 N), which was why we only included references investigating this specific region. However, since the explanation is true in general for SMP clouds in the Arctic, we have modified the paragraph so that it is not only focused on latitudes > 80 and the suggested references have been added. In the paragraph after this one, we focus on the high Arctic region and stick to the corresponding references.

Line 59: Consider linking these two paragraphs to point out that cloud microphysical properties and thus their radiative effect is for example impacted by aerosols. ... (then go into introducing aerosols).

This has been changed now.

Line 62: There are actually a number of studies that have shown sensitivity of Arctic sea ice clouds to aerosols (e.g., Solomon et al., 2018, Stevens et al., 2018, Eirund et al., 2019).Âă

The introduction has now been changed and this sentence has been modified.

Line 65: Also here there are more studies (e.g. the studies mentioned above) that have shown a strong impact of CCN changes to the radiative balance at the surface.Âă

As we have changed the introduction this sentence is now removed.

Line 88: "indirectly inferred" - this is very vague, what exactly did they show?Âă

We have changed this. It is now: However, these analyses were not performed for the high Arctic and they did not explicitly investigate the relation between Aitken particles

and cloud properties or cloud sustenance. Instead, they focused on the correlation between aerosol particles and cloud droplets."

Line 89: From this section it is not clear to me what is different south of the ice edge as compared to over pack ice and between the Arctic and the high Arctic. Please be more specific in what previous studies have shown for which conditions and why more research is necessary.Âă

We have now written about these differences and added corresponding reference.

Line 102: This is confusing to me. According to Koehler theory, large particles should always dominate the CCN availability as they activate more easily. Maybe change "even at low total aerosol concentrations" to "low accumulation mode aerosol concentrations"?Âă

We agree with the reviewer that the sentence was unclear. We have changed it to: "We initialize the models with a range of aerosol size distributions and explore if Aitken mode particles can help sustain the cloud or if only accumulation mode aerosols control cloud properties (i.e. cloud droplet, rain and ice mixing ratios), even at low accumulation mode concentrations."

Methods:Âă

Line 121: Are both of these models commonly used for simulating Arctic mixed-phase clouds? I have read several studies including MIMICA, but to me the RAMS model is rather uncommon for simulating Arctic clouds, so it would be good to add some references here that have used these models previously for similar studies.Âă

We have added references for both models. However, even if one of the models had not been specifically designed/used for Arctic mixed-phase clouds we still believe that it would be of interest to show that it generates the same qualitative results as a model that has been used to simulate these clouds.

Line 168: Here is one example (in addition to the introduction), where you point out the

importance aerosol-cloud interactions for the surface energy budget, which is however never addressed. If you mention it as it is done here, it needs to be analyzed, otherwise please remove.Âǎ

This is a good point. In the new version, we have addressed the importance of Aitken mode particles on radiative fluxes.

Line 170: According to the beginning of this section I assume the radiosonde was launched over sea ice, such that the surface conditions in the model are set to sea ice? If this is the case, please explicitly mention.Âǎ

Yes, the surface conditions were set to sea ice. We have now stated that explicitly, both here and in the Simulation setup section. This sentence is now: "During the ice drift, radiosondes were launched from the ice surface every 6 h and provided profiles of thermodynamic properties (e.g., pressure, temperature, relative humidity) and wind speeds (cf. Figure 1)."

Line 173: "The cloud base and cloud top were nearly constant during the cloud lifetime (500 and 1000 m, respectively)." This implies to me that the cloud has been observed over a longer time period, however, in Figure 1 it looks like the observations show only one point in time. Please clarify and/or change the layout of Figure 1 (see my comment regarding Figure 1).

Yes, the cloud was observed over a longer time period as stated in the section 2.2.: "To investigate a case with a quasi-steady-state cloud regime, the simulations are based on a period that was characterized by a persistent, low-level SMP cloud observed from 18 UTC 30 August to 12 UTC 31 August 2008.". The observations in Figure 1 (now Figure 2) do not show only one point in time, which was also mentioned in the figure caption: "The retrieved values of LWP and IWP for the observed period are shown as 25th, 50th (median) and 75th percentiles.". However, to be more clear, we have modified the caption of Fig. 2:" The retrieved values of LWP and IWP for the observed cloudy period defined in section 2.2 and up to the height of the model domains are

shown as 25th, 50th (median) and 75th percentiles."

Line 196: "The concentrations are chosen to cover typical aerosol size distributions often encountered in the summertime high Arctic (Heintzenberg and Leck, 2012; Leck and Svensson, 2015)." – for both, accumulation and Aitken mode aerosols?Âǎ

Yes, for both accumulation and Aitken mode aerosols. The cited studies show the aerosol size distributions in the high Arctic, i.e. include both modes.

Line 197: Better use "assumed" rather than "considered"?Âǎ

"considered" has been changed with "defined".

Line 200: As you performed quite a large set of simulations, if would be helpful for the reader to have an additional table including all simulations and the varying type/number of aerosol and model that could be referred to here. Also simulation AC20_AK20 could be clearly marked as baseline or control simulation.Âǎ

We also thought that a table was a good idea, however, once we made it we did not think it was particularly useful (we have included it as a supplement, just FYI). Therefore, it is not included in the new version of the manuscript.

Line 203: "fluxes were small" – how small (please be specific)? Is a prescribed flux of zero justified?Âǎ

The observed fluxes were usually smaller than 5 W m-2 during the ASCOS ice drift period with probability peaks around zero (Tjernström et al., 2012). Previous modelling studies simulating the ASCOS case (e.g., Stevens et al., 2018) have also used zero surface fluxes. We have added this explanation in the new version of the manuscript.

Same line: Is the choice of 0.844 for surface albedo arbitrary or is there a reference?Âǎ

We have now added "(cf. Sedlar et al., 2011)" as the reference for the surface albedo value.

Same paragraph: Was there large-scale advection? And how was the roughness length defined? Also I assume the surface condition was set to sea ice? Please add these information to the simulation setup.

There is no large-scale advection and it is now stated in the manuscript. The information about the roughness length and sea-ice surface conditions have also been added. Âă Line 210: Based on what conditions was a spin-up of 2 h chosen?Âă

We consider that after 2h a stable cloud is developed. The information has now been added: "The first 2 h are assumed to be a spin-up period and is therefore excluded from the figures and analysis. After the spin-up period, the cloud layer is stable in the baseline simulations".

Line 229: Why were these simulations only performed with MIMICA?Âă

These simulations have now been done also with RAMS. The results are included in the new version of the manuscript (Subsect. 3.3.1, Figures 11 and 12).

Line 232: "relatively ice-free" An ice crystal number concentration of 0 L-1 is completely ice free, not only relatively ice-free. I would remove "relatively".Âă

We agree, "relatively" has been removed.

Results:Âă

Figure 1: As mentioned above, it looks like you are comparing the temporal evolution of the modeled clouds with an observed temporal snapshot (if this is the case). It would be helpful to have more information about the observations (also how are the percentiles derived, are the observations constant in time?) in the text and the Figure caption. Also it would be interesting to know, if the models and the observations cover the same vertical range of the cloud (if these information are available from the observations). You could consider changing the layout of this figure to height on the y-axis and cloud liquid water and ice content over height on the x-axis.

We have changed the figure caption to: "... The retrieved values of LWP and IWP for the observed cloudy period defined in section 2.2 and up to the height of the model domains are shown as 25th, 50th (median) and 75th percentiles..." For more details, please see our reply to the comment line 173. Âă Line 255: It looks to me that RAMS has a higher autoconversion, as also qr is higher (as shown in Figure 2). If this is indeed the case, maybe mention autoconversion as an additional point.Âă

The figure shows qr profiles for the baseline simulation only and indicates slightly more rain in RAMS just below the cloud. The cloud qr values are quite similar among the baseline cases simulated by the two models. This is now explicitly stated: "Rain mixing ratios are similar for the two models (Fig. 3c and 3d), but RAMS produces slightly more rain below the cloud.". Moreover, looking at the figure A4 where the qr profiles are presented for all cases we can conclude that there is in general more rain in the cloud in MIMICA than in RAMS. This is especially noticeable when comparing the cases with low accumulation mode particles where RAMS does not produce a stable cloud. MIMICA simulates a thicker cloud in all cases comparing to RAMS, with more turbulence (now added as Figure A6) and more liquid in general, as a consequence of 2-3x higher cooling rates at the top of the cloud (as is stated in the manuscript).

Line 259: Could another reason be the inclusion of hail in RAMS? Does hail maybe increase surface precipitation, which then reduces the overall LWP and IWP in RAMS? Have you tried to switch hail off?Âă

The surface precipitation is in general higher in MIMICA than in RAMS, which agrees with higher cloud qr values produced by MIMICA (the surface precipitation figure is now included in the manuscript, Figure A7). We have also checked the individual contribution from rain, ice and hail to the surface precipitation in RAMS (not shown). The contribution from hail is 2 orders of magnitude smaller than the one from rain and 1 order of magnitude smaller than the one from ice; thus the inclusion of hail in RAMS does not change the overall conclusions. This is now discussed in the manuscript.

Line 265: Is it possible that the additional rain formation in RAMS stabilizes the cloud and prevents a continuous cloud top rise as seen in MIMICA? Âă

As we pointed out above, the cases in MIMICA generally have more rain within the cloud and more surface precipitation than the ones in RAMS (Figure A4, as well as added Fig. A7). Based on all analyses we have performed we think that the thicker clouds in MIMICA (with rising cloud tops, stronger turbulence and more liquid in general) are most likely the consequence of the different cloud top cooling rates produced by the two models.

Line 266: Where do these peaks in qi come from? I would expect to see corresponding peaks for example in the radiative cooling rate, which would hint towards enhanced growth by deposition at certain times, but I cannot find any evidence there. Âă

The peaks in total ice mixing ratio in MIMICA come from different collection rates of rain by graupel that arise with time (now added as Figure A8). The change in collection rates with time is a consequence of peaks in updrafts and downdrafts (the same applies for all cases, figures are shown for the baseline simulation only (Figure A8)). This has been also added in the new version of the manuscript: "Both models also simulate similar values of total ice mixing ratios, although MIMICA produces a few stronger vertical bands after 6 h of simulation (Fig. 3e and 3f). This type of pronounced bands is a result of strong collection rates of rain drops by graupel, which appear at different times due to different temporal distributions of updrafts and downdrafts (Figure A8).".

Line 270: Entrainment can also lead to drying (Ackermann et al., 2004) - was there a moist layer present in the observations? Maybe it would be good to show the initial profiles as measured by the radiosonde to see the temperature and specific humidity (or total moisture) vertical distributions? ÂăAlso is looks like there is enhanced evaporation of cloud droplets just above cloud top (Figure 3a,b), which would hint towards a drier layer overlying the cloud? In this case entrainment would rather dry than moisten the cloud. Âă

We agree that the entrainment may also lead to drying depending on the humidity above the cloud. Stratocumulus-topped boundary layers in the high Arctic are usually capped by both temperature and humidity inversions leading to the entrainment of moist air into the boundary layers. This was stated in the introduction in the previous version. However, we agree with the reviewer that we could point out more clearly that this also applies to our simulations. We have included a figure with the radiosonde observations of absolute temperature, potential temperature and specific humidity (Figure 1) where both the temperature and humidity inversions can be observed. The sentence has also been changed to: "The reason is most likely the higher entrainment rates at cloud top in MIMICA (not shown) that bring more water vapor into the cloud from the moist air that is present across the humidity inversion, which caps the cloud-topped boundary layer (Figure 1c)." There is indeed a thin layer of cloud droplet evaporation just above the cloud. However, based on entrainment rates and the humidity profiles (both observed and simulated) we expect positive vapor flux into the cloud layer.

Line 278: Is the correlation of condensation with updrafts shown anywhere? Or is this a general statement? In the latter case please add a reference. Âă

This conclusion was drawn from an analysis of the RAMS results and it is thus not a general statement. We have clarified this by adding "(not shown)".

Figure 3: Please consider changing the colorscale. I understand that smaller values in MIMICA have to be represented, but for example in Figure 3d or f I cannot see the total magnitude at all. Âă

This has been changed, we have now added a colorscale for each plot.

Line 329: "thins" rather than "shrinks" Âă

We have changed "shrinks" with "thins".

Line 334 and Figures 5 and A2: Is this the cooling or the heating rate (such that neg. values indicate a cooling as in Brooks et al. (2017) which is what I assume following

your arguments)? Also the differing colorscales are very confusing, especially since the numbers are very small and an additional zero can easily been overseen. Please consider changing the units to K/h or K/d and try to keep the colorscales the same. As the cloud top is essential for the analysis, it might be worth zooming into the cloud top to better resolve the magnitude of the cooling there. You could also add cloud top such as shown in Figure A3 within Figure 5, so the reader can easily identify cooling at the cloud top.Âǎ

Yes, the figures represent the heating and not the cooling rates. This was a mistake. We thank the reviewer for all suggestions regarding these figures, they have now been changed accordingly (now Fig. 6 and A2).

Line 337: In line 253 you say that stronger radiative cooling rates in MIMICA produce a higher LWP, while here you state that the higher qc leads to stronger cooling in MIMICA. Of course it's a feedback where high LWP changes the cloud emissivity, which in turn increases longwave cooling which thenÂǎagain favors enhanced turbulence and condensation. However, above a certain threshold of approximately 40 g m-2 the sensitivity of longwave cooling to LWP becomes small (e.g. Garret and Zhao, 2006) which is why I assume you point out the difference in cloud top liquid water here. Please be more specific in your line of argumentation of the qc/cloud top cooling feedback and also refer to other studies of Arctic mixed-phase clouds which have investigated this correlation as well.Âǎ

We agree with this point. We do not consider that the difference in the radiative cooling rates between the two models arise from this feedback, it is simply a consequence of the different radiation schemes that the models use. The additional tests we performed and presented in the Appendix support this conclusion. In the new manuscript version, we have explicitly written that the discrepancy in cloud top cooling does not arise from the correlation between LWP and cloud top emissivity and we have cited Garret and Zhao (2006). "For the baseline case, both models simulate a relatively thick cloud (LWP > 40 g m-2), which indicates that the differences in the cloud top cooling rates between

MIMICA and RAMS do not arise from the difference in simulated LWP (cf., Garrett and Zhao, 2006)." is now added.

Line 343: Not only the entrainment of moist air, but also the increase in vertical motions as a result of more turbulence can favor condensation and maintain cloud liquid (Shupe et al., 2008).Âǎ

This sentence has been removed from the new version for other reasons, but we do agree with this point.

Line 347: As mentioned in my point 3, does maybe the different vertical resolution also play a role here as well?Âǎ

Please see our reply to point 3.

Figure 4: It looks like the high end of the colorscale is never reached, thus consider adjusting it. Âǎ This has been adjusted.

Line 380: I see a very strong signal of Aitken mode particles for low number concentrations of accumulation mode particles, which underlines the main message of the manuscript (cloud maintenance for 6 h). Why do you say there no clear trend?Âǎ

We do think there is an influence of Aitken mode particles in all cases in both models as it was stated in the first sentence of the paragraph:" Figure 6 shows that adding Aitken mode particles generally increases the amount of cloud droplet water in both models, i.e. the particles serve as CCN and allow formation of additional cloud droplets.". The student t-test confirms it. What we wanted to say is that there is no clear trend in the Aitken mode influence between the RAMS cases with different (and low) accumulation mode concentration, i.e. that is not clear whether the Aitken particles are more important for the cases with acc=0 cm-3, acc=3cm-3 or acc=5 cm-3. We agree with the reviewer that the sentence was confusing. Moreover, we now think the figure shows that the influence of Aitken mode particles in RAMS generally also becomes less pronounced as the accumulation mode concentration increases (as it is in MIMICA). This

is now changed in the manuscript:" In both models the influence of smaller particles on cloud droplet mixing ratio thus generally decreases with increasing accumulation mode concentration (Fig. 7). Nevertheless, the differences in cloud droplet mixing ratio are statistically significantÂăfor all pairs of different Aitken mode concentrations in both models, except for the MIMICA pair AC20_AK200 and AC20_AK20 (according to a student t-test with a 95 % confidence level on the time averages in the cloud layer). In other words, both models show that Aitken mode particles have a significant impact on the cloud droplet mixing ratio, at least up to 20 cm-3 of accumulation mode particles in RAMS and at least up to 10 cm-3 in MIMICA.".

Line 387: Here you say there is a significant impact, while above you write there is no clear trend (see my previous comment). I would agree with this statement and emphasize it more, as it is one of the main messages of the manuscript.Âă

Please see our reply above.

Line 410: why do more Aitken mode particles lead to more turbulence? Because of more latent heat release through condensation? If this is the case, please clearly state it. However, the updrafts for low accumulation mode and different Aitken mode particles look very similar, but this could be due to temporal averaging I assume.Âă

In the previous version this statement was written as a possible explanation: "Despite the higher number of cloud droplets, an increase in the Aitken mode particle concentration may lead to stronger turbulence". In the new version, we have added a figure with the cloud-averaged time-mean TKE values (Fig. A6). The figure shows that there is indeed more turbulence in the cases with increased number of Aitken mode particles.

Line 415/Figure A9: I assume Figure A9 shows spatial and temporal statistics? In this case I cannot see a temporal evolution in the updrafts that would determine the temporal evolution of rain. Please make this argument clearer.Âă

Yes, the statistics were done based on a 20min period for all supersaturated grid boxes

in the model domains. We do realize the time variability in updrafts could not been seen from the figure. We refer now to another figure, which has been added in the new version (Fig. A8).

Line 418: Previous studies have also shown an impact of CCN on cloud ice and increased LW cooling of CCN-perturbed clouds has been identified as driving force for increased immersion freezing and growth by deposition (Possner et al., 2017, Solomon et al., 2018, Eirund et al., 2019). As the ice crystal number concentrations is fixed, immersion freezing does not play a role here, but I would suspect that LW cooling does also impact growth by deposition and thus qi. This might be worth exploring/mentioning.Âă

We agree with the reviewer. We have now explained the relationship between increased LW cooling and the influence of CCN on ice: "The influence of Aitken mode particles on ice is in general larger in MIMICA than in RAMS consistent with the stronger cloud top cooling rates simulated by MIMICA that favors the ice formation through immersion freezing and growth by vapor deposition when the number of CCN increases (Possner et al. 2017; Solomon et al., 2018; Eirund et al., 2019).".

Line 426: This is an interesting finding. The effect of CCN changes for different background ice crystal number concentrations/INP concentrations has been studied previously (Stevens et al., 2018, Possner et al., 2017). There, also a smaller CCN impact was found for higher ice crystal number concentrations /INPs, which would agree with your findings and might be worth mentioning.Âă

This has been added now: "This result agrees well with previous studies that have investigated the influence of CCN in mixed-phase clouds with different background INP or ice crystal concentrations (e.g., Possner et al., 2017; Stevens et al., 2018).".

Supersaturation statistics:Âă

Line 487: Why did you choose a 20 min interval around 6 h of simulation time? From Figure 6 it looks like strongest signals in qc are either in the beginning or towards the

end of the simulations (in RAMS); is your choice of time period linked to any criteria? Ăă

In the RAMS cases with low accumulation mode concentrations (acc=0,3,5 cm-3), the signals are stronger at the beginning than towards the end due to the cloud dissipation after 6h. However, we consider that 6h, as the middle of the simulation period, can be more representative for the statistics than some earlier times. If we chose a time towards the end of the simulation then there would not be a cloud in the all RAMS cases. The period of 20min is chosen to cover data variability and thus be more representative for the statistics than the output from one time step.

Line 515: Again, previous studies have also shown an increase in cloud-top radiative cooling in seeded clouds (see my previous comment). Please cite previous studies accordingly. Ăă

We have now cited previous studies.

Qualitative comparison of model results with observational data for the High Arctic: Ăă

Line 338/Section 5: This section is very interesting and gives the authors the opportunity to make statements about the relevance of their work in a broader scope. However, this section is relatively short in my opinion and could be expanded. Especially, it would be interesting to know when/where is including Aitken mode particles important and should be considered in modeling studies? Also the authors could expand section 5 to a general discussion of the results and could compare their findings to previous work which investigated the response of Arctic cloud to varying CCN concentrations. Ăă

We have extended this section to a general discussion and made two subsections, 5.1 Qualitative comparison of model results with observational data for the High Arctic and 5.2. General importance of Aitken mode particles for low-level mixed-phase cloud properties. In the later one, we have discussed the spatial and seasonal importance of smaller particles.

Summary and conclusions: Ăă

Line 580: "Aitken mode aerosols have a significant impact on the cloud droplet amount" - All Figures show differences in mixing rations, not number concentrations (hence cloud droplets could also be larger, not necessarily more numerous). Did I miss something or should this be "cloud liquid water"? I agree that the increase in qc most likely results from an increase in Ndrop, but this has not been shown, as far as I can see. Ăă

We thank the reviewer for pointing this out. We have now changed it.

Line 582: Again "higher number of cloud droplets", same comment as above. Ăă

It is changed.

Please also note the supplement to this comment:
https://acp.copernicus.org/preprints/acp-2020-665/acp-2020-665-AC1-supplement.pdf

———————————————————

| aerosol mode | accumulation | Aitken |
| --- | --- | --- |
| particle number concentration (cm$^{-3}$) | 0
3
5
10
20 | 20
200 |

**Fig. 1.** Table with simulation setup

---

## Author Comment (AC2) · 2 Nov 2020

General comments:

The authors employed two numerical models to perform a series of simulations to study

the importance of Aitken mode aerosol particles for cloud sustenance in the summer-time high Arctic. The messages in the abstract seem to be clear. After reading through the main text of the paper, I find some interesting results. But I am also confused by arbitrary model and simulation configurations and overwhelmed by poorly interpreted, disorganized, and probably unnecessary results. Throughout the manuscript, the authors used a lot of speculations in their reasoning where solid evidences are expected. The writing also has huge room for improvement. I listed my major and minor concerns as well as suggestions regarding the technical aspects of the manuscript below.

We thank the referee for his/her comments that have helped us improve and clarify the manuscript.

Both models utilized in the study have previously been used for simulating Arctic mixed-phase clouds, which is now explicitly stated in the manuscript (Sect. 2.1). We have on purpose used the default configurations of the two models, as this is what typically would be used for e.g. a model-observation comparison/evaluation.

We have answered the comments by the reviewer in a point-by-point fashion, revised the manuscript accordingly and made a thorough effort to provide a clearer and more organized interpretation of the results.

One of the co-authors on the study is an English native speaker who has carefully read the manuscript and paid attention to linguistic mistakes.

Specific comments (major):

If I understand correctly, the main idea of this paper is that, for some combinations of model configurations (i.e., Aitken mode and accumulation mode aerosol number concentration, aerosol kappa value, and ice number concentration), the modeled cloud can survive through 12 h, meaning that during this period, the clouds can maintain sufficient liquid water against processes that depletes liquid water (like subsidence, entrainment of warm air, losing moisture due to glaciation and precipitation, so on) and

generate enough supersaturation to activate the prescribed Aitken mode aerosol. Was the specified aerosol size distribution for each simulation used from the beginning of the simulation? How important are aerosols during the spin-up? In other words, are the differences among simulations using the same model and between MIMICA and RAMS due to the activation behavior when the turbulent motion is very weak? What if all simulations (in each model) begin with a robust cloud and fully-developed the turbulence, spun-up using same configurations, and then switch to different aerosol number concentrations? I don't think a juicy but non-turbulent cloud is a realistic starting point to test aerosols' impacts on sustaining clouds or produce results that are relevant to the real world. Please justify this choice.

Both models use an initial cloud droplet number concentration of 30 cm-3. This is now clarified in the manuscript (Sect. 2.3). In other words, both models start with a fully developed cloud, which thereafter is maintained (or not) based on the prescribed aerosol size distributions and modelled supersaturations. We do believe this type of model setup is realistic as it roughly represents conditions where a cloud forms close to the marginal sea ice zone and thereafter is advected in over the sea ice (where aerosol sources are either absent or small).

A few other questions related to initialization and spin-up: Is the initial cloud size distribution related to the Aitken mode and accumulation mode aerosol? For each model, are the liquid water content profiles in all simulations identical at the beginning? Are all microphysical processes (e.g., all processes related to precipitations) turned on from the beginning?

The profile of cloud water mixing ratio is the same in both models at the beginning of all simulations as it is stated in the manuscript. All microphysical processes are active at the beginning of the simulations, which is now also stated (please see Sect. 2.3).

It is worthwhile to be more specific about the activation of aerosol particles in MIMICA and RAMS as configured for this study. It seems that the activation scheme in MIMICA

is identical to the one used to generate Fig. A10. What about in RAMS? Are the ss in Fig. 10 and Fig. 11 same as those used in the activation scheme in the models?

Figure A10 (now Fig. A12) shows theoretical critical supersaturation values as a function of particle dry diameter for a range of kappa values, where kappa is a measure of the hygroscopicity of a multi-component or single-component aerosol particle (Petters and Kreidenweis, 2007). The activation schemes in both models use this theory for cloud condensation nuclei activation as stated in the Sect. 2.1. The reason why look-up tables are used in RAMS is only due to computational efficiency. The supersaturation (ss) statistics shown in Fig. 10 and Fig. 11 (now Fig. 14 and Fig. 15) are directly derived from the model output, i.e. the ss values shown are indeed used to activate aerosols in the two models. The critical dry diameters shown in the figures are calculated for the ss values (i.e. for 75th and 99th ss percentiles) obtained from the models, based on the relationship presented in figure A12 and explained in detail in Petters and Kreidenweis (2007). We have not made any changes in the manuscript as all this information was already available.

It seems that the authors tried to use an observed sounding to set up the baseline simulations, and then perform sensitivity tests on top of that. However, the authors did not provide enough details (for example, dedicated figures) for readers to understand the case. Please consider showing some details. I found the ASCOS sounding available from https://bolin.su.se/data/ascos-radiosoundings. Did the authors use the original 0535 UTC 31 August 2008 sounding from this archive? Or an idealized version of it? The authors used the observed CCN to justify the use of AC20_AK20 as the baseline simulation. However, the cloud layer in the aforementioned sounding seems to be decoupled from the surface. If this is the case, does it still make sense to use surface measurement to determine the base case? The authors mentioned a few times that Arctic stratocumulus may entrain moist air from above the cloud top, but provided no evidence. (Whether other studies showed entrainment of moisture from above the cloud could happen is irrelevant to this study.) Initial sounding (together with profiles

from the middle of the simulations) can be used to show whether there is moist layer above the cloud top.

We agree with the reviewer that it is useful to show the radiosonde observations and have therefore added the new Figure 1. We have indeed used the original soundings from 0535 UTC 31 August 2008, as is stated in the manuscript. In Figure 1, we also display the simulated profiles after 6h of simulation. Showing the initial profiles does not make sense as these are the same as the radiosonde observations.

There were no other CCN measurements available from the ASCOS campaign apart from those obtained from a CCN counter that was situated on Oden. However, we do agree with the reviewer that the CCN concentrations could be different within (or just below) the cloud due to the decoupled boundary layer. The sentence: "Since the surface boundary layer typically was decoupled from the turbulent layer associated with a cloud (Tjernström et al., 2012), it is however not certain if the CCN concentrations measured at the ship were representative for the cloud layer (cf. also observed vertical profiles of particle concentrations in Igel et al., 2017, Figure 1)." has been added in Section 2.2.

Initial profiles of absolute temperature, potential temperature and specific humidity (Figure 1) show that the simulated case was characterized by both a temperature and humidity inversion.

Choice of model configurations, shared simulation setup, and experiment design are perplexing and arbitrary. Why were the vertical resolutions different, especially as the authors suspected that the different vertical resolutions may be the source of some discrepancies between the simulation results from the two models (e.g., L254). Why were the microphysics in RAMS so much more complicated (even with hail turned on) than in MIMICA? Why was aggregation turned off due to low ice number concentration, but why only turned off for MIMICA? Why was MIMICA used for ice number concentration sensitivity test while certain ice-related budget terms are not available from it (L289)?

As we briefly mentioned above, we wanted to use the models in their standard config-
urations and examine the similarities/differences in the simulated results. The default
setup in RAMS is a fixed vertical grid spacing while in MIMICA the default grid spacing
is variable. We agree that the difference in grid spacing could be a source of dis-
crepancies and have therefore run an additional MIMICA baseline simulation with a
fixed vertical grid spacing of 10 m as in RAMS. The test did not show any significant
differences in the simulated cloud microphysical properties compared to the MIMICA
baseline case with a variable grid spacing. This is now explained in the new version of
the manuscript (Sect. 2.3).

It is not completely clear to us why the reviewer considers that the microphysics in
RAMS is much more complicated compared to MIMICA. We agree that the inclusion of
hail could change the microphysical rates for e.g. a deep convective cloud, but for an
Arctic stratocumulus cloud the hail production is in general very small. Indeed, in all
RAMS simulations 97.9-99.7% of the ice is present as ice crystals so the riming treat-
ment plays a minor role in the simulations. We have also checked the contribution of
hail to the surface precipitation. It is 2 orders of magnitude smaller than the contribution
from rain. This information is now added in the manuscript (please see Sect. 2.3). The
aggregation process is actually switched on in both models, the statement that it was
turned off in MIMICA was a mistake and has been corrected.

We agree with the reviewer that the ice tests could have been performed with both
models and have now made the corresponding simulations. We present results from
both MIMICA and RAMS in the new version of the manuscript and we find that the
results are consistent.

Much of the results regarding rain and ice are only superficially described and dis-
cussed, with no obvious connection to the main goal of the paper, and sometimes
contain errors. A few examples are provided here. Printer-friendly version Discussion
paper L277: "The pockets of condensation and evaporation present in the main cloud
layer are well-correlated with updrafts and downdrafts and they tend to cancel each

other in the mean. This is why the average condensation rate in the main cloud is of the same order of magnitude as the one in the sub-cloud layer." Totally lost.

We have clarified the sentence. RAMS produces a sub-cloud condensation layer (as Fig. 4b shows) and the sentence explains why this layer is present and so pronounced. The relatively high mean value is a result of infrequent but strong condensation due to sub-cloud convection simulated in RAMS.

L415: "The presence of both positive and negative differences with time is a result of differences in cloud dynamics with different distributions of updrafts and downdrafts with time that govern the rain production in the cloud (cf. Fig. A9)." But there is nothing about "distributions with time" in Fig. A9.

We agree with the reviewer that the Fig. A9 (Now Fig. A11) was not a good choice to refer to. We now instead refer to Fig. A8, which shows that there are different distributions of updrafts and downdrafts with time (Fig. A8b,c). The figure also shows that the temporal evolution of updrafts and downdrafts is well correlated with the evolution of the collection rate of rain drops by graupel (Fig. A8a) and therefore also with the rain budget. Results are only presented for the baseline simulation, but the results are qualitatively the same for all simulations.

L421: "change in the Aitken mode particle number concentration results in that maximum updrafts are reached at somewhat different times", what is the significance of this?

The sentence has been rephrased in the new version of the manuscript.

L422: "Differences are in general greater in MIMICA than in RAMS since there is a slightly more total ice in MIMICA". Does "since" mean "because" here? Does greater total ice water content (or path?) have to lead to greater differences? Other than the "entrainment of moisture aloft", here are a few additional examples of unacceptable speculation.

[Figure]

We have now changed the reasoning for the observed difference in the Aitken mode influence on total ice mixing ratio between the two models. Previous studies have shown that a pronounced CCN influence on ice in CCN-perturbed clouds can be related to stronger LW cloud top cooling (e.g., Possner et al., 2017, Solomon et al., 2018, Eirund et al., 2019), which is present in MIMICA and not in RAMS. Please see the Subsection 3.3.1 for more details.

L288-289, "Examining the total ice deposition/sublimation rates would most likely lead to similar rates between the two models". (BTW, "The ice crystal deposition and sublimation rates are higher in RAMS than in MIMICA since the two models partition the total ice deposition differently among ice hydrometeor categories" is also just speculation, isn't it?)

The sentence in the brackets is not a speculation. We have indeed analysed the (available) deposition rates for all ice species in both models and concluded that they differ between the same ice hydrometeor categories in the two models. The sentence has been reformulated to:" The ice crystal deposition and sublimation rates are higher in RAMS than in MIMICA since the two models partition the total ice deposition differently among ice hydrometeor categories (not shown).". "Examining the total ice deposition/sublimation rates would most likely lead to similar rates between the two models" is on the other hand something that we cannot be sure about and we therefore removed this sentence from the new version of the manuscript.

L411-412: "an increase in the Aitken mode particle concentration may lead to stronger turbulence and more cloud liquid water production". Turbulence intensity and cloud liquid water budget can be diagnosed. It does not make sense to speculate ("may lead to").

This is a good suggestion and we have now included a figure of the time-mean, resolved TKE within the cloud layer (Fig. A6). In both models, the TKE is higher in cases with a higher number of Aitken mode particles (comparing the pairs of simulations with

the same number of accumulation mode particles). The relation between the cloud-averaged TKE and cloud water mixing ratio is obtained by comparing Fig. 5 and Fig. A6. The text has been changed accordingly.

Minor comments:

Contents in Sections 2.1 and 2.3 are not clearly separately. For example, why are number of model grid points and resolution introduced in Section 2.1 but domain size in Section 2.3?

We agree that it is more suitable to provide this information in the Simulation setup section and have moved it there. Otherwise, we believe that the contents of Sections 2.1 and 2.3 are clearly separated.

Please clarify a few model or simulation configurations issues: Is it correct that the time step is 2 s for MIMICA? What about RAMS? The terminal speed is only introduced for MIMICA, what about RAMS? What are the references for the terminal speed formulas used? Is the 0.2 L-1 ice number concentration based on ASCOS observations? Need reference. Why is divergence set to 1.5E-6 s-1? Is the same value used for all model layers?

It is correct that the time step in MIMICA is ∼2s as stated in the manuscript. In RAMS, the time step is 1.5s and this is now explained (Sect. 2.1).

Information about the terminal fall speed in RAMS as well as the references for both models have also been added (Sect. 2.1).

We have added the Stevens et al. (2018) study as a reference for the ice crystal number concentration (Sect. 2.3).

In both models, the large-scale divergence is constant in the whole model domain (at all model levels) and the value is chosen to produce a stable cloud layer (cf. Stevens et al., 2018). The explanation together with the reference has now been added in the text (Sect. 2.3).
L130: Is the wet deposition calculated for any tracers in either model?

No, this has not been included as stated in the Section 2.3 (i.e. the aerosol size distributions are prescribed).

L346: Since they are simplified LW cooling calculation that only depends on LWC, why not using same formula for both MIMICA and RAMS?

As mentioned before, we wanted to use models with their standard configurations. Using two different models allows us to evaluate if the results are dependent on the details of a specific model or if we can draw more general conclusions. The additional tests that we have done with e.g. the simplified LW cooling calculations (in MIMICA and RAMS) or a fixed vertical grid spacing (in MIMICA) are meant to indicate which differences in the model configurations lead to the discrepancy in the simulated results. We think this information could be useful for the future modelling studies.

L349: "Stable cloud base", do you mean "steady"?

This sentence has now been removed from other reasons.

L383: "Fig. 5", should be Fig. 6?

Yes, this was a mistake in typing. However, this paragraph has been modified in the new version of the manuscript and the sentence has been removed.

Examples of poor writing: L265: "total ice", do you mean "total ice mixing ratio"? There are a few "total ice" throughout the text.

We agree that "total ice mixing ratio" is more precise than "total ice" and have changed it in that place. However, we did not find any other example of incorrect use of the "total ice" term throughout the text. For instance, in the sentence: "The ice crystal deposition and sublimation rates are higher in RAMS than in MIMICA since the two models partition the total ice deposition differently among ice hydrometeor categories (not shown)." or in the sentence: "Examining the total ice deposition/sublimation rates

would most likely lead to similar rates between the two models, but these rates are not available in MIMICA", the term "total ice" is used as a possessive adjective and not as a noun. The models partition the (total) ice differently and have different (total) ice deposition/sublimation rates. The simulated (total) ice is then expressed with a mixing ratio variable.

L377: "available water vapor" what is this?

We have modified the sentence to: "A higher number of cloud droplets decreases the maximum supersaturation and the amount of water vapor in the cloud available for activation of smaller particles."

Suggestions on technical issues:

Please consider marking cloud top and base whenever relevant.

We have added the cloud top and cloud base heights in all figures where it could be done.

Colors are saturated in some figures, e.g., Fig. 3f. Please adjust.

We have adjusted it.

L292: "from the different model descriptions", should be "configurations"?

"model descriptions" has been changed with "model configurations".

Please also note the supplement to this comment:
https://acp.copernicus.org/preprints/acp-2020-665/acp-2020-665-AC2-supplement.pdf

---

## Referee Report (RR1)

Review of **"The importance of Aitken mode aerosol particles for cloud sustenance in the summertime high Arctic: A simulation study supported by observational data"** by Bulatovic et al.

**General comment**

The authors have substantially improved the manuscript and have included the reviewers' suggestions. The discussion has been expanded and now better embeds the study in the existing literature. Similarly, the introduction has been restructured and additional relevant studies are cited. All figures have been improved according to the suggestions. I still have a few minor comments which should be addressed prior to final publication, which are listed below.

**Specific comments**

Line 46: "cloud liquid growth" should rather be "cloud liquid water increase"

Line 47, line 50: "water vapor" instead of "vapor"

Line 274: How are IWP and LWP defined (i.e. including or excluding precipitation)?

Line 293: consider changing "is better" with "performs better"

Line 300: RAMS also produces vertical bands of increased rain throughout the cloud, e.g. at hours 8 and 9. Can you explain those? Again, is the autoconversion similar for both models? Or is the autoconversion rate especially high in RAMS? You show the condensational growth of raindrops, but (if available) it could be interesting to have a look at the conversion rate from cloud droplets to rain drops.

Figure 3: Thanks very much for including the cloud top and base height. However, how did you calculate cloud top and base height? Please add this information.

Line 367: "the cloud base altitude change**s**…"

Line 411: "when no accumulation mode particles **are** present"

Line 439: "vary" instead of "very"

Line 452: "in the two model**s**"

Line 456 ff: I appreciate the additional discussion of the radiative budget in the simulations. Did you also analyze the impact of SW radiation at the surface (as done in Figure 10 for LW radiation)? I would assume that especially (and most likely only) in summer there might also be an impact. Or did you only investigate SW radiation at the model top, as written in line 461?

Also, did you look at the **net** surface LW and SW radiation? Overall it would be interesting to see, how the **net** surface energy balance changes for the inclusion of Aitken mode particles and how these results could be extrapolated to the summertime high Arctic radiative budget.

Line 681: Apart from the choice of prescribed/prognostic ICNC/INPs, secondary ice formation may also play a relevant role in determining the cloud evolution, especially in summertime Arctic MPCs (Sotiropoulou et al., 2020; doi: 10.5194/acp-20-1301-2020) and at the temperatures shown in Figure 1. However, this is not mentioned at all. Are secondary ice processes omitted (in the first version of the manuscript I read they are)? Does this also introduce additional uncertainties or effects (i.e. a

potentially higher ice fraction and a decreased influence of Aitken mode particles)? This could also be mentioned in e.g. Section 5.2.

---

## Author Response (AR2)

Review of "**The importance of Aitken mode aerosol particles for cloud sustenance in the summertime high Arctic: A simulation study supported by observational data**" by Bulatovic et al.

General comment

The authors have substantially improved the manuscript and have included the reviewers' suggestions. The discussion has been expanded and now better embeds the study in the existing literature. Similarly, the introduction has been restructured and additional relevant studies are cited. All figures have been improved according to the suggestions. I still have a few minor comments which should be addressed prior to final publication, which are listed below.

We thank the reviewer again for his/her careful reading and further comments.

Specific comments

Line 46: "cloud liquid growth" should rather be "cloud liquid water increase"

The sentence is now:" The turbulence further increases cloud liquid water as strong overturning means strong updrafts that allow efficient condensation of water vapor onto cloud droplets."

Line 47, line 50: "water vapor" instead of "vapor"

"vapor" has been changed with "water vapor".

Line 274: How are IWP and LWP defined (i.e. including or excluding precipitation)?

In both models the LWP and IWP include precipitation by definition. The observed LWP and IWP do also include precipitation. This has now been added in the caption of the Figure 2.

Line 293: consider changing "is better" with "performs better"

It is now "performs better".

Line 300: RAMS also produces vertical bands of increased rain throughout the cloud, e.g. at hours 8 and 9. Can you explain those? Again, is the autoconversion similar for both models? Or is the autoconversion rate especially high in RAMS? You show the condensational growth of raindrops, but (if available) it could be interesting to have a look at the conversion rate from cloud droplets to rain drops.

In RAMS, the available diagnostic rate is the sum of the autoconversion and the collection of cloud droplets by rain drops. This sum is higher in RAMS compared to MIMICA for the baseline simulation (please see attached figure). However, we cannot know whether the autoconversion rate or the collection rate has a stronger impact on the rain budget in RAMS and which rate contributes to the vertical bands observed in the qr profiles simulated by this model.
In the new version of the manuscript we have now added a sentence: "For the baseline simulations, the sum of the autoconversion rate and the collection rate of cloud droplets by rain drops is two orders of magnitude higher in RAMS than in MIMICA, which contributes to the higher rain mass mixing ratios in the RAMS sub-cloud layer.".

[Figure]

Figure 3: Thanks very much for including the cloud top and base height. However, how did you calculate cloud top and base height? Please add this information.

In both models, the cloud layer is considered to be present between altitudes where the cloud water mixing ratio (qc) is larger than $1 \cdot 10^{-6}$ kg/kg. This has now been added in the text: "In both models, the cloud base (cloud top) height is the altitude above (below) which the cloud droplet mixing ratio exceeds the value $1 \cdot 10^{-6}$ kg/kg.".

Line 367: "the cloud base altitude changes…"

This has been changed.

Line 411: "when no accumulation mode particles **are** present"

"are" has been added.

Line 439: "vary" instead of "very"

This has been corrected.

Line 452: "in the two model**s**"

"s" is added.

Line 456 ff: I appreciate the additional discussion of the radiative budget in the simulations. Did you also analyze the impact of SW radiation at the surface (as done in Figure 10 for LW radiation)? I would assume that especially (and most likely only) in summer there might also be an impact. Or did you only investigate SW radiation at the model top, as written in line 461? Also, did you look at the **net** surface LW and SW radiation? Overall it would be interesting to see, how the **net** surface energy balance changes for the inclusion of Aitken mode particles and how these results could be extrapolated to the summertime high Arctic radiative budget.

We thank the reviewer for this point. The reason why we have investigated the Aitken mode influence on the top of the model (TOM) SW radiation only is that we have considered that the cloud impact (i.e. different concentrations of Aitken mode particles) would be the most pronounced on the reflected component of the SW radiation. We have now also investigated the Aitken mode influence on (net) SW radiation at the surface and it is *not* significant.

The sentence has been changed to: "Both models simulate no significant influence of Aitken mode particles on the SW radiation, consistent with the low insolation (not shown).", i.e. we have removed the part "at the top of the model domains".
We have also looked at the net LW surface radiation fluxes and they show the same result as the corresponding downward fluxes. The net flux is the difference between downward and upward flux, and the later one only depends on the surface emissivity and the surface temperature - and these two parameters are prescribed in both models so they do not vary between the simulations.

Line 681: Apart from the choice of prescribed/prognostic ICNC/INPs, secondary ice formation may also play a relevant role in determining the cloud evolution, especially in summertime Arctic MPCs (Sotiropoulou et al., 2020; doi: 10.5194/acp-20-1301-2020) and at the temperatures shown in Figure 1. However, this is not mentioned at all. Are secondary ice processes omitted (in the first version of the manuscript I read they are)? Does this also introduce additional uncertainties or effects (i.e. a potentially higher ice fraction and a decreased influence of Aitken mode particles)? This could also be mentioned in e.g. Section 5.2.

This is a good point. Secondary ice processes are not considered in the present study and could obviously have an impact on the total ice fraction simulated by the model. In the section 5.2., the last sentence is now: "The results most likely depend on whether the ice crystal concentrations are prognostic or prescribed and if secondary ice processes are considered in the calculations of the ice crystal number concentration (Sotiropoulou et al., 2020).".